# Remote carbon cycle changes are overlooked impacts of land-cover and land management changes

Suqi Guo[1], Felix Havermann[1], Steven Johan De Hertog[2,3], Fei Luo[4,5,6], Iris Manola[4], Thomas Raddatz[7], Hongmei Li[7,8], Wim Thiery[2], Quentin Lejeune[9], Carl-Friedrich Schleussner[9,10,11], David Wårlind[12], Lars Nieradzik[12], Julia Pongratz[1,7]

[1]Department of Geography, Ludwig-Maximilians-Universität München, Munich, Germany
[2]Department of Water and Climate, Vrije Universiteit Brussel, Brussels, Belgium
[3]Q-ForestLab, Department of Environment, Universiteit Gent, Ghent, Belgium
[4]Institute for Environmental studies, Vrije Universiteit Amsterdam, Amsterdam, the Netherlands
[5]Royal Netherlands Meteorological Institute (KNMI), De Bilt, the Netherlands
[6]Centre for Climate Research Singapore (CCRS), Singapore, Singapore
[7]Max Planck Institute for Meteorology, Hamburg, Germany
[8]Helmholtz-Zentrum Hereon, Geesthacht, Germany
[9]Climate Analytics, Berlin, Germany
[10]Integrative Research Institute on Transformations of Human-Environment Systems (IRI THESys), Humboldt-Universität zu Berlin, Berlin, Germany
[11]Geography Department, Humboldt-Universität zu Berlin, Berlin, Germany
[12]Department of Physical Geography and Ecosystem Science, Lund University, Lund, Sweden

*Correspondence to*: Suqi Guo (Suqi.Guo@geographie.uni-muenchen.de)

**Abstract.** Land-cover and land management changes (LCLMCs) have a substantial impact on the global carbon budget and, consequently, global climate via the biogeochemical (BGC) effects. The commonly considered BGC effects refer to the direct influence of LCLMCs on local carbon stocks ("local BGC effects"). However, LCLMCs also influence climate by altering the local surface energy balance due to changes in land surface properties such as albedo, leaf area, and roughness ("local biogeophysical (BGP) effects"). Altered local air mass properties can impact regions remote from LCLMCs through advection and changes in large-scale circulation ("nonlocal BGP effects"). Previous studies have shown potentially substantial nonlocal BGP effects on temperature and precipitation. Given that the terrestrial carbon cycle strongly depends on climate conditions, this raises the question of whether LCLMCs can trigger remote carbon cycle changes ("nonlocal BGC effects") - a currently overlooked potentially large climate and ecosystem impact. To assess the nonlocal BGC effects, we analyze sensitivity simulations for three selected types of hypothetical large-scale LCLMCs: global cropland expansion, global cropland expansion with irrigation, and global afforestation, which were performed by three state-of-the-art Earth system models. We separate the nonlocal BGC effect using a checkerboard-like LCLMC perturbation that has previously only been applied to BGP effects. We show that nonlocal BGC effects on vegetation and soil carbon pools persistently accumulate, exceeding natural fluctuations and typically becoming detectable within the first 40 years after LCLMCs. By the end of our 160-year simulation period, nonlocal BGC effects lead to an absolute magnitude of change in total terrestrial carbon stock by 1 to 37 GtC, with strong changes over the densely forested Amazon region (0.2 to 7 GtC) and central Congo Basin region (0.3 to 15 GtC), depending on models and LCLMCs implemented. For the irrigation scenario, the nonlocal BGC effects are comparable to the total BGC effects with the nonlocal-to-total ratio for vegetation carbon pools commonly reaching around 90%. Our

results reveal that the nonlocal BGC effects could be substantial and call for these effects to be considered for accurate impact assessment and sound policymaking. This becomes even more relevant when LCLMCs are expected to play a pivotal role in achieving the Paris Agreement's goal of limiting global warming below 1.5 °C above pre-industrial levels.

## 1 Introduction

Land-use-induced land-cover and land management changes (LCLMCs) alter climate by greenhouse gas (GHG) emissions and removals as well as by affecting the surface energy balance, which are summarized as biogeochemical (BGC) and biogeophysical (BGP) effects, respectively (Bonan, 2008; Boysen et al., 2020; Bright et al., 2017; Pongratz et al., 2021). As a key strategy to mitigate climate change, LCLMCs play an important role for the Paris Agreement's goal to limit global warming below 1.5 °C above pre-industrial levels (Grassi et al., 2017; Jia et al., 2019; Roe et al., 2021). LCLMCs also support other sustainable development goals (SDGs), such as zero hunger (goal 2) or life on land (goal 15) (Hurlbert et al., 2019). To optimize LCLMCs as strategies to mitigate climate change and pursue win-win solutions with other SDGs, a comprehensive and deep understanding of the LCLMCs' climate effect is required.

LCLMCs influence the local climate via energy, water, and momentum fluxes due to changed land surface properties such as albedo, leaf area, and roughness. These direct consequences are collectively known as the local BGP effect (see Table 1). Observational data by design quantify the local BGP effects (Bright et al., 2017; Duveiller et al., 2018) and this effect can also be isolated from Earth system model simulations (Kumar et al., 2013; Malyshev et al., 2015; Winckler et al., 2017a), as explained below. Studies reveal, for example, a regionally distinct pattern with warming related to deforestation in the tropics as well as much of the temperate regions and a cooling effect in the high latitudes (Duveiller et al., 2018; Mahmood et al., 2014; Winckler et al., 2019b), with species-dependent variation (Bright et al., 2017). Local BGP effects can be substantial, with regional annual mean temperature changes of several degrees Celsius, as shown for changing a forest to grassland (Bright et al., 2017; De Hertog et al., 2023; Winckler et al., 2017a).

However, LCLMCs also influence remote climate via advection of the altered air mass properties and possible changes in large-scale circulation, namely the nonlocal BGP effects (see Table 1; see also Laguë & Swann, 2016; Portmann et al., 2022; Winckler et al., 2019a). The nonlocal effects can only be quantified by models. The definition of local vs nonlocal scales depends on the application – while changes in micro- or mesoscale phenomena could be resolved by high-resolution modelling, our study focuses on global impacts of large-scale LCLMCs connected to synoptic scales. Studies changing forest to grasslands show that idealized deforestation, while it can warm the climate on a global average with local BGP effects, brings about nonlocal BGP effects that cool the climate by several tenths of a degree on global average (Winckler et al., 2019a). This cooling effect dominates the overall climate impact and is consistent across most models after historical deforestation (Winckler et al., 2019a). Meier et al., 2021 show that substantial nonlocal effects in precipitation are caused by afforestation. In Europe, these changes often exceed 0.1 mm d$^{-1}$ and are at least comparable to the local effects and in some regions even exceed the local effect. Other studies investigating land management, suggesting that nonlocal effects may be strong: Irrigation, for example,

has been found to change precipitation and temperature (Gormley-Gallagher et al., 2022; Hirsch et al., 2017; Thiery et al., 2017, 2020) even in regions unaffected by the application of irrigation (Cook et al., 2015; De Vrese et al., 2016; Mahmood et al., 2014). Regionally, the nonlocal irrigation effects can dominate the precipitation change with a magnitude of several tenths of mm d$^{-1}$ (De Hertog et al., 2024). The nonlocal irrigation effect on temperature is notable too, with a magnitude of several tenths of a degree Celsius (De Hertog et al., 2023; De Vrese et al., 2016), depending on models and scenarios (De Hertog et al., 2023), particularly the implemented area extent (Sacks et al., 2009).

LCLMCs influence climate substantially also via BGC effects: In the period 2010-2019 LCLMCs emissions account for 25 % of total anthropogenic GHG emissions (Hong et al., 2021), or 10-15 % if only $CO_2$ emissions are considered (Friedlingstein et al., 2023). Moreover, pre-industrial LCLMC $CO_2$ emissions contribute about one third to the current cumulative emissions leading to one fourth of today's higher temperatures (Pongratz & Caldeira, 2012). However, research mainly concentrates on the direct LCLMCs effect on climate (see Table 1): The carbon (C) emissions and removals at the location of the LCLMCs as a result, for example, of the clearing of carbon-dense forests for agricultural lands, regrowth of natural vegetation when agricultural areas are abandoned, or altered carbon stocks due to a management practice. However, BGC cycles and C pools also strongly depend on environmental conditions. The rise in atmospheric $CO_2$ concentration over the industrial era has turned the land's soil and vegetation to a substantial carbon sink, absorbing one quarter to one third of the current anthropogenic $CO_2$ emissions (Friedlingstein et al., 2023), in response to the overall beneficial effects of $CO_2$ on plant growth (Walker et al., 2021). Changes in climate can increase or decrease C stocks, such as warming in boreal regions extending the growing season, or increased droughts and fires reducing carbon stocks. Overall, the climate effects have offset the natural land sink by about 20 % in the last decade (Friedlingstein et al., 2023). The underlying processes, besides disturbances, are the strong dependence of plants on temperature, moisture, and other BGP drivers. Given that nonlocal BGP effects may be large, as described above, it becomes obvious that LCLMCs may not only impact remote regions' climate discernibly, but also their C stocks. Agriculture, forestry, and natural ecosystems may be affected, and any changes in C stocks will feedback on global climate change by altering the atmospheric $CO_2$ concentration. Despite these potentially severe consequences, research has not yet addressed this indirect effect of LCLMCs before.

Earth system models show significant variability in their results of climate and carbon cycle changes due to differing implementations of LCLMCs, vegetation processes, and parameterizations (Boisier et al., 2012; Boysen et al., 2020; Fisher & Koven, 2020). This leads to substantial divergence in the magnitude and even the sign of LCLMC-induced BGP effects (De Hertog et al., 2023; Pongratz et al., 2021). Few studies have compared hydrological responses, revealing regional precipitation changes that also diverge in sign (Boysen et al., 2020; De Hertog et al., 2024; Pitman et al., 2009). To address model uncertainty, employing multiple models is a common strategy (Eyring et al., 2016; Jia et al., 2019). Previous studies, using a multi-model approach, mainly focus on total BGP effects (Boisier et al., 2012; de Noblet-Ducoudré et al., 2012; Pitman et al., 2009, Yao et al. in prep.), yet inter-model comparisons of nonlocal BGP effects and certain LCLMCs like irrigation remain scarce (De Hertog et al., 2023, 2024; Pongratz et al., 2018). For instance, in CMIP6 simulations, only three Earth system models included irrigation (Al-Yaari et al., 2022).

Here, for the first time, we analyze simulations with three state-of-the-art ESMs combined with irrigation schemes to address the impacts of the nonlocal BGP effects on terrestrial C stocks (called "nonlocal BGC effects" from now on; see Table 1) due to LCLMCs. We present a method to quantify the nonlocal BGC effects using ESMs and apply this method to three selected types of LCLMCs: cropland expansion without irrigation, cropland expansion with irrigation, and afforestation. We investigate these effects under present-day climate conditions, as they are of greatest relevance to near-term decisions on how to use our land. Nonetheless, our approach is fully transferable to any scenario with different climate conditions. More specific aims of our study are (i) to quantify the simulated global development and spatial distribution of nonlocal effects of LCLMCs on different terrestrial carbon pools (Sects. 3.1 and 3.2), (ii) to assess the importance of nonlocal BGC effects in relation to the total effects, which consist of both local and nonlocal BGC effects and represent the overall carbon cycle response at the location of the LCLMCs (Sect. 3.3), (iii) the point in time when the nonlocal BGC effects become larger than the natural internal variability (Sect. 3.4), and (iv) the sensitivity of nonlocal BGC effects to temperature and soil moisture (Sect. 3.5). This work thus forms the basis for expanding our understanding of the unintended side-effects of LCLMCs, including in regions where no LCLMC occur. This work also presents an approach for quantifying unintended $CO_2$ emissions or removals in remote areas. When assessing the overall climate benefit of an LCLMCs practice, this remote carbon cycle response needs to be accounted for.

**Table 1: Definitions of Land-Cover and Land-Management Change (LCLMC) Effects (BGP: Biogeophysical, BGC: Biogeochemical).**

| LCLMCs effects | Affected regions | Definition |
|---|---|---|
| Local BGP effects | Regions with LCLMCs | LCLMCs influence the local climate via energy, water, and momentum fluxes due to changed land surface properties such as albedo, leaf area, and roughness. |
| Nonlocal BGP effects | Regions without LCLMCs | LCLMCs influence remote climate via advection of the altered air mass properties and possible changes in large-scale circulation. |
| Local BGC effects | Regions with LCLMCs | LCLMCs directly influence the local carbon stocks by changing the local vegetation type or its management. |
| Nonlocal BGC effects | Regions without LCLMCs | LCLMCs influence remote carbon stocks through climate changes driven by nonlocal BGP effects. |

## 2 Methods

### 2.1 Earth system models setup and scenarios

The ESMs and scenarios used in our study are summarized here, with full detail provided in De Hertog et al. (2023). Three state-of-the-art ESMs were included in this study: the Community Earth System Model (CESM) version 2 (Danabasoglu et al.,

2020), the Max Planck Institute Earth System Model (MPI-ESM) version 1.2 (Mauritsen et al., 2019), and the European Community Earth System Model (EC-EARTH) version EC-Earth3-Veg (v3.3.3.1; Döscher et al., 2022). Our model versions and spatial resolutions are identical to the CMIP6 setup with dynamically coupled model components of land, atmosphere, and ocean. Consistency with CMIP6 has the advantage that the models have been evaluated and shown to be generally in line with

the historical climate evolution (Craigmile & Guttorp, 2023; Danabasoglu et al., 2020; Fan et al., 2020; Rashid, 2021; Wehner et al., 2020). Further, our results complement and can be directly compared to analyses based on CMIP6 and spin-off projects like the land-use model intercomparison project (LUMIP, Lawrence et al., 2016), which are driven by other land-use changes or climate forcings.

We analyzed ESM output of three idealized LCLMCs scenarios: cropland expansion without irrigation (CROP), cropland

expansion with irrigation (IRR), and afforestation (FRST), as well as one control scenario without any LCLMCs as a reference (see Table 1). The general idea of all scenarios is not to present plausible realizations under realistic socio-economic pathways, but to simulate large-scale LCLMCs. This has two advantages: First, simulating disturbances at large scale will increase the signal-to-noise ratio. While the idealized nature of the scenarios prohibits conclusions for concrete realizations of future global LCLMCs, the higher signal-to-noise ratio allows us to better establish the potential importance of nonlocal BGC effects and

provide a proof of concept to account for them. Second, unlike historical LCLMCs or realistic future scenarios where LCLMCs occur in limited regions, idealized global LCLMCs enable the estimation and comparison of impacts across most regions worldwide.

**Table 2: Overview of the Earth system model scenarios analyzed in our study, together with a brief description of the simulated land-cover and land management changes (PFT: plant functional type).**

| Scenario name | Land cover change | Land management change |
|---|---|---|
| Control (CTL) | None. Constant land cover of the year 2014. | No wood harvest and irrigation |
| Cropland expansion without irrigation (CROP) | We replace all PFTs that are neither cropland nor bare soil (such as pasture, grassland, shrubland, and forest) in the 2014 land cover with crop PFTs. This ensures cropland occupies 100% of the hospitable land within a grid cell for MPI-ESM and CESM. For EC-Earth, however, the cover fraction also depends on climate conditions. Performed on half of the land grid cells in a checkerboard pattern. | No wood harvest and irrigation. |
| Cropland expansion with irrigation (IRR) | Same as in the CROP scenario. Performed on half of the land grid cells in a checkerboard pattern. | Irrigation on all cropland PFTs. |
| Afforestation (FRST) | We replace all PFTs that are neither forest nor bare soil (such as pasture, grassland, shrubland, and forest) in the 2014 land cover with forest PFTs. This ensures forest | No wood harvest and irrigation. |

| | occupies 100% of the hospitable land within a grid cell for MPI-ESM and CESM. For EC-Earth, however, the cover fraction also depends on climate conditions. Performed on half of the land grid cells in a checkerboard pattern. | |


All scenarios are branched from the official CMIP6 historical concentration-driven simulation at the end of the year 2014 with a simulation period of 160 years. The general idea behind these choices is to derive LCLMCs effects under approximately present-day climate, to be independent of scenario choices and to be indicative for land-use choices that could be taken today, and to run simulations that are sufficiently long to average out internal variability. The scenarios are forced using the same

anthropogenic forcing (trace gas, troposphere anthropogenic aerosols, and population density) and natural forcing (solar radiation, wildfire, lightning, and natural stratosphere aerosols) of the year 2014. The only forcing that differs among the scenarios is the prescribed land cover change and land management change (see Table 1). For the control scenario we use a constant year 2014 land-use data set from the end of the CMIP6 historical scenario (originating from the LUH2 dataset (Hurtt et al., 2020)), but without any land management implemented, i.e., no irrigation and no wood harvest. In the CROP and the

FRST scenario, we applied a land cover change to crop or forest plant functional types (PFTs) for the entire hospitable land of a grid cell (see Appendix A for more details). This was done for half of all land grid cells (Fig. C1). We chose to change grid cells such that the final land mask has a checkerboard pattern of changed and unchanged land cover (see Fig. C1 for LCLMC distributions in this study; see Winckler et al. (2017a) for illustration on checkerboard approach). By this homogeneous distribution of changed and unchanged grid cells we could apply an established method to separate local and nonlocal effects

of LCLMCs (see Sect. 2.2 for more details). The distribution of the specific crop or forest PFTs (e.g., tropical broadleaf evergreen forest or tropical deciduous forest) remains constant in the changed grid cells (see Appendix A for more details). The IRR scenario uses the CROP scenario and additionally applies each model's native irrigation scheme to all LCLMC grid cells globally (see Appendix B for more details).

Despite adopting the identical experimental design, ESMs are diverse in LCLMCs implementations (see Table 2). We

summarize the main differences; further details are provided in Appendix B. (i) Different from the other two models, EC-Earth uses the dynamic vegetation model LPJ-GUESS, which allows PFTs to compete on six stand types (natural, pasture, urban, crop, irrigated crop, and peatland). Consequently, converted cropland or natural land could be further replaced by other PFTs based on climate conditions, leading to less target land cover. (ii) For the FRST scenario, we could only prescribe the entire natural stand instead of explicit forest in EC-Earth. As a result, depending on the climate, grassland coexists with the forests

and shrubs. (iii) In EC-Earth, the physical properties of trees gradually establish depending on biomass buildup, in contrast to an immediate physical forest representation in MPI-ESM and CESM. (iv) Different from the other two models, in EC-Earth the water cycle components between LPJ-GUESS and the atmospheric model (Integrated Forecasting System, IFS) are not coupled. This implies that irrigation affects the water budget only within LPJ-GUESS, without directly impacting the atmosphere through surface water and energy fluxes (Döscher et al., 2022). However, irrigation within LPJ-GUESS influences

vegetation growth and physical properties (e.g., leaf area index (LAI) and vegetation cover), which subsequently impacts the atmosphere by surface energy exchange.

The global distributions of land-cover changes and magnitude of irrigation application are shown in Fig. C1. Generally, EC-Earth shows smaller changes of land area fractions of forest for the afforestation scenario and, to a lesser extent, also of cropland for the cropland expansion scenario than the other two models (Fig. C1). The amount and spatial distribution of

irrigation varies substantially between all three models. Notably, the approach taken by MPI-ESM shows irrigation in the boreal latitudes, different from the other two models.

**Table 3: Comparison of land-cover and land management changes (LCLMCs) implementations across Earth system models (PFT: plant functional type).**

| LCLMCs implementation | MPI-ESM | CESM | EC-Earth |
|---|---|---|---|
| Land surface scheme and resolution | JSBACH3 (1.88°×1.88°) | CLM5 (0.90°×1.25°) | HTESSEL with LPJ-GUESS vegetation dynamics (0.7°×0.7°) |
| Dynamic vegetation | Dynamic competition among PFTs switched off | Dynamic competition among PFTs switched off | Dynamic competition between all PFTs allowed in each stand type |
| Implementation of afforestation | Uses prescribed transitions to model-intrinsic forest PFTs | Uses prescribed land cover states for model-intrinsic forest PFTs | Does not support exact afforestation fractions; afforestation occurs by expanding natural fraction, allowing coexistence of grass, shrub, and tree PFTs |
| Establishment of plant physical properties | Immediate establishment after land cover change, lower sensitivity of surface properties to state and age of forest | Immediate establishment after land cover change, lower sensitivity of surface properties to state and age of forest | Succession to forest via grass and shrub PFT composition, high sensitivity of surface properties to forest fraction of area |
| Land-atmosphere coupling concerning irrigation | Fully coupled between land and atmosphere, i.e. irrigation impacts the atmosphere directly via the water cycle and indirectly via vegetation changes | Fully coupled between land and atmosphere, i.e. irrigation impacts the atmosphere directly via the water cycle and indirectly via vegetation changes | Irrigation impacts on vegetation type and LAI are simulated in LPJ-GUESS and influence land-atmosphere fluxes; direct water cycle impacts are not represented |

## 2.2 Isolating local and nonlocal LCLMCs effects, including the nonlocal signal in the terrestrial carbon stocks

Isolating the local and nonlocal signal of LCLMCs – though only in terms of biogeophysical effects – has become a common type of analysis in LCLMCs studies. For our analysis, we follow the well-established "checkerboard" approach by Winckler, which has also been applied in previous studies (Winckler et al., 2017a, Winckler et al., 2019, De Hertog et al., 2023). However, no previous study has isolated the nonlocal signal in the terrestrial carbon stocks. We here describe the full approach of isolating local and nonlocal LCLMCs effects, including our new developments concerning the nonlocal signal in the terrestrial carbon stocks:

We simulate LCLMCs by applying the land cover forcing in a checkerboard pattern of changed and unchanged land cover (Fig. C1), as described in Sec. 2.1, and compare it to a reference with only unchanged land cover (CTL). As we are interested in global impacts of large-scale LCLMCs at the scale of 100 km upwards, which matches the resolution of the ESMs, we implement the checkerboard pattern at the native resolution of each ESM (Tab. 3). We then follow the post-processing approach by Winckler et al. (2017a) to separate effects induced by LCLMCs into the local and nonlocal signal.

We assume that the signal in grid cells without LCLMCs consists only of nonlocal effects, while the signal in grid cells with LCLMCs captures total (local plus nonlocal) effects. To separate these effects, we proceed with the following steps: (i) We calculate the difference between the control scenario and the LCLMC scenarios. In grid cells without LCLMCs, the difference is entirely driven by the nonlocal effects. (ii) Nonlocal effects are also present in grid cells with LCLMCs. To obtain the global distribution of nonlocal effects, we spatially interpolate the result of the unchanged grid cells to the changed LCLMCs grid cells by applying a linear interpolation (nearest-neighbor interpolation for coastal land grid cells). (iii) We then calculate local effects in the grid cells with LCLMCs by subtracting the nonlocal effects from the total effects. (iv) To obtain the global distribution of local effects, we spatially interpolate the values from the changed LCLMC grid cells to the unchanged grid cells, using the same interpolation approach as in step (ii).

While Winckler et al. (2017a) applied this method to climate variables such as surface energy balance, as did De Hertog et al. (2023) for the same simulations used in our study, we apply the method to detect the nonlocal BGC effect. Specifically, (i) we subtract the spatially gridded C stocks of the control scenario from the LCLMCs scenarios. This difference at the grid cells without LCLMCs must be entirely driven by the nonlocal BGP effects (Fig. 1b). By contrast, at the grid cells where LCLMCs occur, direct local effects such as the loss of vegetation carbon by replacing forest with cropland co-occur with nonlocal effects (Fig. 1a). (ii) To obtain the global distribution of nonlocal effects, we spatially interpolate the result of the unchanged grid cells to the changed LCLMC grid cells. The result is the globally distributed nonlocal BGC effect due to 50 % global LCLMCs according to the checkerboard pattern.

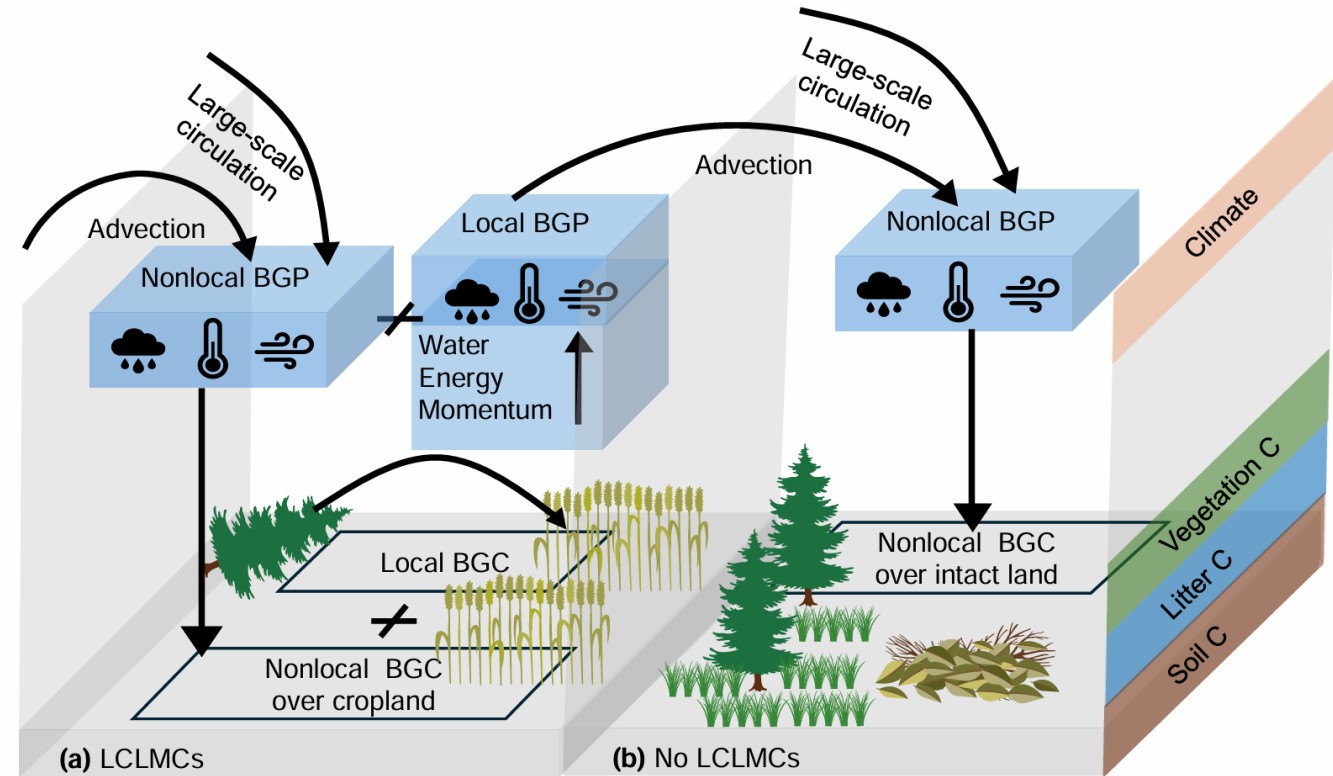

**Figure 1: Local and nonlocal BGP and BGC effects of LCLMCs in two adjacent grid cells in the CROP scenario as an example. In grid box (a) with LCLMCs, both local and nonlocal BGP and BGC effects occur. In grid box (b) without LCLMCs, only nonlocal BGP and BGC effects occur. Local BGP effects describe changes in local climate due to altered energy, water, and momentum fluxes from changed land surface properties. Nonlocal BGP effects result from advection of altered air mass properties and changes in large-scale circulation. Nonlocal BGC effects are carbon cycle responses to nonlocal BGP climate changes, while local BGC effects represent direct carbon emissions and removals induced by local LCLMCs.**

### 2.3 Calculation of the nonlocal to total ratio

To assess the relevance of the nonlocal BGC effects in comparison to the overall changes in carbon induced by LCLMCs, we compute the ratio between nonlocal and total BGC effects, which are comparable across models and scenarios. Therefore, we use the difference between the LCLMC scenarios and the control scenario directly, without any interpolation. For the nonlocal BGC effects, we take only those grid cells without LCLMCs into account. For the total effects, we use all grid cells. This implies that the total effects include both the sum of local and nonlocal effects on grid cells with LCLMCs (Fig. 1a) and the nonlocal effects on grid cells without LCLMCs (Fig. 1b). The magnitude of the total BGC effects therefore refers to the actual simulation signals of a given LCLMC scenario, even though it apply to a highly idealized scenario of checkerboard changes in LCLM as in our case. The magnitude of the nonlocal BGC effects is calculated from the unchanged grid cells only. For all our analyses except for Fig. 5 and 6, we spatially interpolate the nonlocal BGC effects to also estimate the effects over changed grid cells. However, for the calculation of the nonlocal to total ratio, this would have created an inconsistency when comparing to total effects in these grid cells: Interpolation of nonlocal BGC effects from unchanged grid cells (Fig. 1b) to changed

LCLMC grid cells is based on the assumption of similar C stock changes, driven by similar nonlocal BGP effects, between adjacent grid cells ignoring the vegetation types changes due to LCLMCs. By contrast, the nonlocal effects actually simulated

at the changed grid cells are the C stock response with changed LCLMCs (Fig. 1a). For example, in the FRST simulation, the nonlocal BGC effects represent the response of the present-day vegetation types to the BGP climate changes induced by the forest cover increase elsewhere. In contrast, the nonlocal effects occurring over the changed grid cells represent the response of the forest to the nonlocal BGP effects. To avoid this inconsistency of a direct comparison here, we restrict the nonlocal BGC effects to unchanged grid cells. The values for the nonlocal BGC effects assumed in this analysis are thus smaller (around half)

than the nonlocal BGC effects we use in the rest of the analysis but can be interpreted more intuitively as the extent to which C stock changes in unchanged grid cells (Fig. 1b) contribute to the overall changes across all grid cells after LCLMCs.

**2.4 Calculation of the time of emergence**

To analyze the temporal development and identify the year from when nonlocal BGC effects pass the model's internal natural variability, we apply the concept of time of emergence (ToE). The ToE identifies the presence of LCLMCs-induced nonlocal

BGC effects and pinpoints the moment when they become detectable. An early ToE characterizes a relatively large and fast impact on the nonlocal BGC effect. The ToE was frequently used in climate predictions and risk assessments (Abatzoglou et al., 2019; Boysen et al., 2020; Hawkins & Sutton, 2012). Following the criterion from Hawkins and Sutton (2012), we use the following Eq. (1) to calculate the signal-to-noise-ratio; the ToE is the first year in which the signal-to-noise-ratio exceeds 1.

$$\frac{S_t}{N_t} = \frac{\frac{1}{m}\sum_{i=t-\frac{m}{2}}^{t+\frac{m}{2}-1} C_i^{nonlocal}}{\sqrt{\frac{1}{n}\sum_{j=1}^{n}(c_j - \bar{c})^2}} \tag{1}$$

$$c_j = c_j^{ctl} - c_j' \tag{2}$$

$$c_j' = K + A \times e^{-\frac{j}{\tau}} \tag{3}$$

At each grid cell, we define the signal ($S_t$) as the 16-year (m) moving mean of the nonlocal BGC effect ($C_i^{nonlocal}$) and the noise ($N_t$) as variability of the detrended control simulation signal ($c_j$, where the index j refers to years of the simulation) for 160 years (n), where $\bar{c}$ is the 160-years mean of $c_j$. Note that capital C refers to the effects, as difference between two

simulations, while lower-case c refers to individual simulations. Since the control simulation is not affected by any changes in forcing, we use it to quantify the internal variability that occurs naturally. However, because of the slow response of the C cycle, the C pools of the control simulation ($c_j^{ctl}$) continued to change after the cessation of anthropogenic alterations in the year 2014 (moving from historical climate, $CO_2$ and land-use changes to constant present-day forcing). To nevertheless derive an approximate value of the internal variability, we apply Eq. (2) to eliminate the long-term trend. Since the evolution of

vegetation and soil C stocks towards their equilibrium value can be approximated by a decaying exponential function, we use Eq. (3) as a fit for $c_j'$, with coefficients $K$, $A$, and $\tau$ determined by the evolution of carbon pools over time.

**2.5 Attribution of nonlocal vegetation C and soil C effects to temperature and soil moisture**

Generally, nonlocal BGC effects arise as a result of climate change (nonlocal BGP effects) and the corresponding response of the terrestrial ecosystems. The sensitivity of this response is governed by various plant physiological processes, including carbon assimilation and plant respiration, while the specific vegetation biomass density can additionally enhance the impact. To better understand which aspects of the nonlocal BGP changes drive the nonlocal BGC effects we apply a multiple linear regression analysis (Eq. (4)), with which we attribute the nonlocal BGC effects to temperature and soil moisture (Franklin et al., 2016; Friedlingstein et al., 2006). We selected two specific factors: near-surface air temperature (called "temperature" from now on) (Fig. C2) and moisture of the upper 10 cm soil layer (called "soil moisture" from now on) (Fig. C3) as explanatory variables since they were more indicative for changes as compared to other similar variables (not shown; tested for JSBACH). By Eq. (4) we estimate the nonlocal C stock change. The regression coefficients of the multiple linear regression serve as indicators of ecosystem sensitivity to temperature and soil moisture.

$$\Delta C_t^{nonlocal} = K_0 + K_1 \times tas_t^{nonlocal} + K_2 \times mrsos_t^{nonlocal} + R \quad \text{(t=1 to 160)} \tag{4}$$

$$\Delta C_t^{nonlocal} = C_t^{nonlocal} - C_{t-1}^{nonlocal} \text{, where } C_0^{nonlocal} = 0 \tag{5}$$

$$C_j^{nonlocal} = \sum_{t=1}^{j} \Delta C_t^{nonlocal} \tag{6}$$

To accurately assess the interannual increment in nonlocal BGC, we estimate the year-by-year difference in annual nonlocal BGC effects, denoted as $\Delta C_t^{nonlocal}$ (see Eq. (5)). $K_0$ denotes a constant, $K_1$ and $K_2$ denote the sensitivity of the carbon cycle to annual-mean temperature ($tas_t^{nonlocal}$) and soil moisture ($mrsos_t^{nonlocal}$), respectively. R denotes the residuals. Compared to the nonlocal BGC effect ($C_j^{nonlocal}$), $\Delta C_t^{nonlocal}$ is influenced less by previous years' climate change and thus has a better correlation with the nonlocal BGP effects of that year. The cumulative change in nonlocal BGC of year j ($C_j^{nonlocal}$) is the sum of annual nonlocal BGC ($\Delta C_t^{nonlocal}$) across the time span before year j (Eq. (6)).

The nonlocal BGP effects of temperature and soil moisture diverge in magnitude and even sign (see Fig. C2 and C3). For the CROP scenario, MPI-ESM presents minor drying and warming in the northern hemisphere high latitudes while CESM and EC-Earth present cooling and wetting, with CESM being more pronounced; MPI-ESM and CESM present warming and drying in most areas of the western Amazon and central Congo Basin while EC-Earth presents warming and wetting; these nonlocal soil moisture discrepancies can be attributed to strong mesoscale effects in EC-Earth, better resolved with high spatial resolution (De Hertog et al., 2024). For the FRST scenario, CESM presents major drying in the northern hemisphere high latitudes, while MPI-ESM and EC-Earth present minor drying. For the IRR scenario, both MPI-ESM and CESM present global cooling and wetting, with cooling being more substantial in MPI-ESM; EC-Earth, however, presents minor warming and drying; these less pronounced nonlocal BGP effects in EC-Earth, compared to the other two models, arise because the water cycle is not fully coupled between the land and atmosphere, blocking the direct impact of irrigation on surface energy fluxes (Döscher et al., 2022). Apart from the IRR scenario, nonlocal soil moisture changes in EC-Earth are typically an order of magnitude smaller than in the other models for the CROP and FRST scenarios.

## 3 Results

First, we analyse the global-integral, transient carbon stocks changes in terrestrial carbon pools (Sect. 3.1). We then concentrate on particular carbon pools that influence total terrestrial carbon stock (cLand) changes as components. Specifically, we analyse changes in vegetation carbon stock (cVeg) and soil carbon stock (cSoil). Litter carbon stocks (cLitter) changes often present similar but generally minor changes compared to cSoil and are thus not shown. We also investigate the magnitude and importance of nonlocal BGC effects from both spatial distribution and relative magnitude perspectives (Sects. 3.2 and 3.3). Next, we investigate when these signals are established over time (Sect. 3.4). Lastly, we attribute the nonlocal BGC effects to climate factors, with climate distributions presented in Figs. C2 and C3 (Sect. 3.5).

### 3.1 Nonlocal effect on global carbon stock changes

Over the 160-year simulation, nonlocal carbon changes accumulate and show saturating trends in some pools across models and scenarios (Fig. 2). Toward the end of the simulation, it is not clear if the natural ecosystem has stabilized or will continue to change under LCLMC-induced nonlocal climate changes.

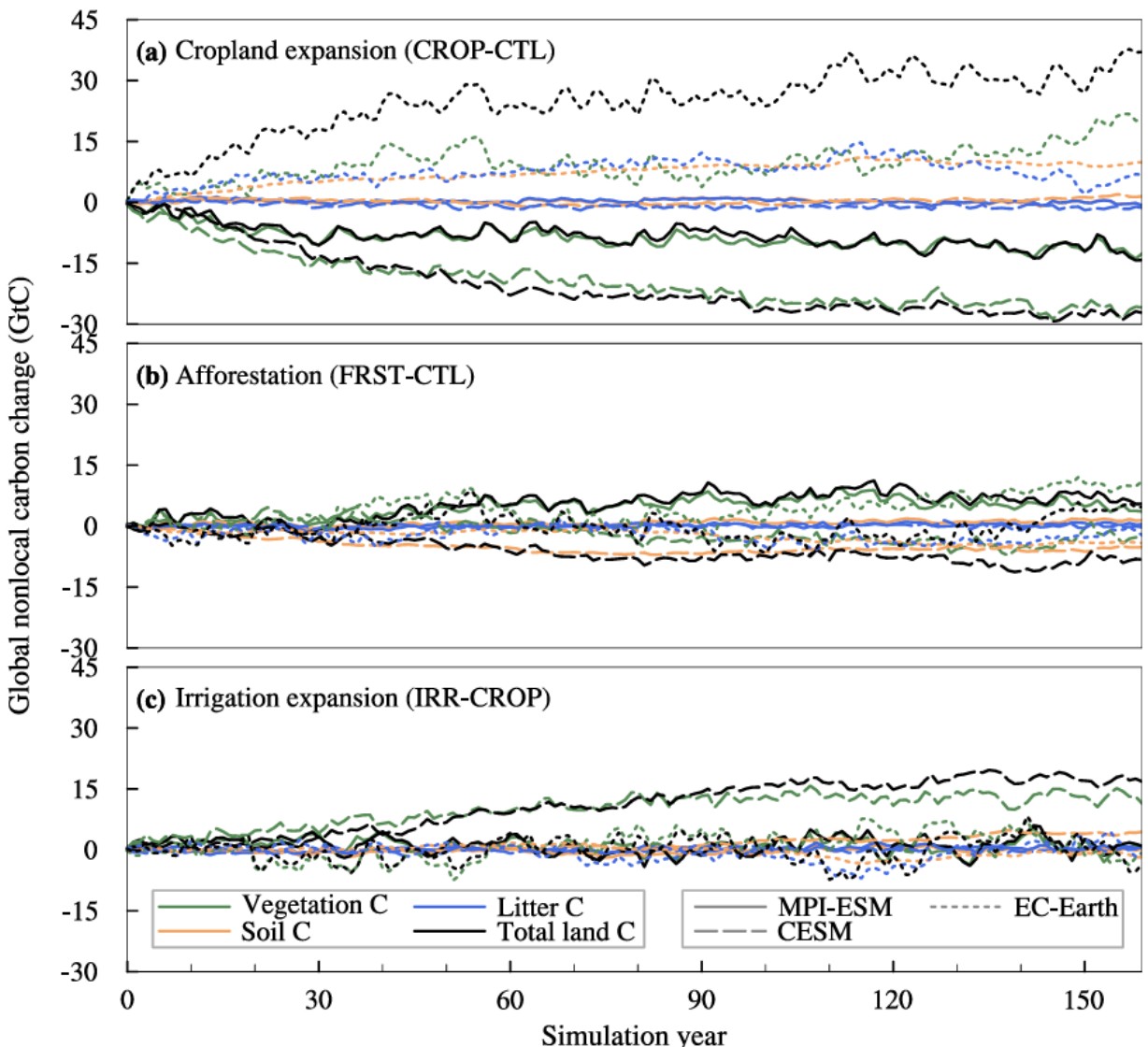

**Figure 2: Simulated nonlocal effect on the development of global terrestrial carbon pools after an idealized change of 50 % of all grid cells: (a) cropland expansion (b) afforestation (c) irrigation of cropland expansion. The total land carbon (black) is separated into vegetation (green), soil (orange), and litter (blue) pools. The results for MPI-ESM (solid lines), CESM (dashed lines), and EC-Earth (dotted lines) are shown for each carbon pool and for total land carbon.**

For the CROP scenario, the global nonlocal cLand is simulated to decrease by -11 GtC in MPI-ESM and -28 GtC in CESM, respectively, on average over the last 30 years of our 160-year simulation period (Fig. 2a). In contrast, EC-Earth simulates a gain of +32 GtC for cLand. For MPI-ESM and CESM, the nonlocal cVeg changes dominate the nonlocal cLand changes, whereas for EC-Earth, also the cSoil and cLitter stocks contribute substantially. The global integral of cSoil and cLitter presents only minor changes over time in MPI-ESM and CESM, but this hides substantial, but opposing, signals among regions.

Additionally, the nonlocal BGC stock changes show strong interannual variability, particularly in MPI-ESM and EC-Earth. This variability is primarily driven by the western Amazon region, while the central Congo Basin shows much weaker variability (see Appendix D for details). This variability is related to the internal climate variability, which varies across regions (Loughran et al., 2021).

In the IRR scenario, CESM shows a substantial increase of +13 GtC in cVeg and a +4 GtC increase in cSoil leading to an overall growth of +18 GtC in cLand (Fig. 2c). In contrast, MPI-ESM and EC-Earth simulate small nonlocal BGC stock gains and losses due to the offset among regions with opposing signals (see Sect. 3.2).

For the FRST scenario, MPI-ESM and EC-Earth simulate cLand increases of +7 GtC and +2 GtC, respectively (see Fig. 2b). In MPI-ESM and EC-Earth, these cLand changes are dominated by cVeg with a growth of +5 GtC and +9 GtC, respectively,

whereas for EC-Earth, both, cLitter and cSoil presents decreasing nonlocal BGC stock effects. Conversely, the results of CESM show a cLand decrease of -9 GtC, mainly due to a decrease of cSoil by -6 GtC and cVeg by -4 GtC. Compared to the CROP scenario, the response of cLand in the FRST scenario starts with a delay after the start of the simulations in MPI-ESM and CESM.

There are several reasons for the divergence across models regarding the magnitude, trend and variability of global-integral,

transient nonlocal BGC effects. First, that BGC effects differ concerning their magnitude divergence occurs in some key regions where nonlocal BGP effects diverge considerably (see Sect. 3.2). For example, for the CROP scenario, opposing nonlocal cVeg and cLand changes between MPI-ESM/CESM and EC-Earth are mainly caused by opposing cVeg changes in the western Amazon region due to opposing nonlocal climate conditions (see Sect. 3.2.1, Sect. 3.5 and Appendix D for details). Additionally, model divergence concerning trend and variability is related to how each model's land surface scheme handles

LCLMCs (see Sect. 2.1). For example, for the FRST scenario, the dynamic vegetation competition and replacement, as well as the gradual establishment of tree physical properties induce oscillations in EC-Earth between gains and losses in nonlocal carbon pools during the simulation period.

## 3.2 The spatial distribution of nonlocal carbon stock changes

### 3.2.1 Nonlocal vegetation carbon stock changes

For the CROP scenario, the spatial distribution of the nonlocal cVeg changes shows a general decrease in the C stock with similar patterns between CESM and MPI-ESM (Fig. 3a, b). However, for EC-Earth, the lower latitude (30º S-30º N) regions show increasing nonlocal cVeg stocks. Especially the substantial cVeg increase in the western Amazon and central Congo Basin regions contrasts with the patterns observed in MPI-ESM and CESM (Fig. 3c). In the low latitudes (17º S-17º N), MPI-ESM and CESM simulates a total loss in cVeg of -12 GtC and -10 GtC, respectively, whereas EC-Earth simulates a gain in

cVeg of +18 GtC. The CESM results additionally show cVeg losses in the Northern Hemisphere high latitudes (41º N-90º N) of -15 GtC.

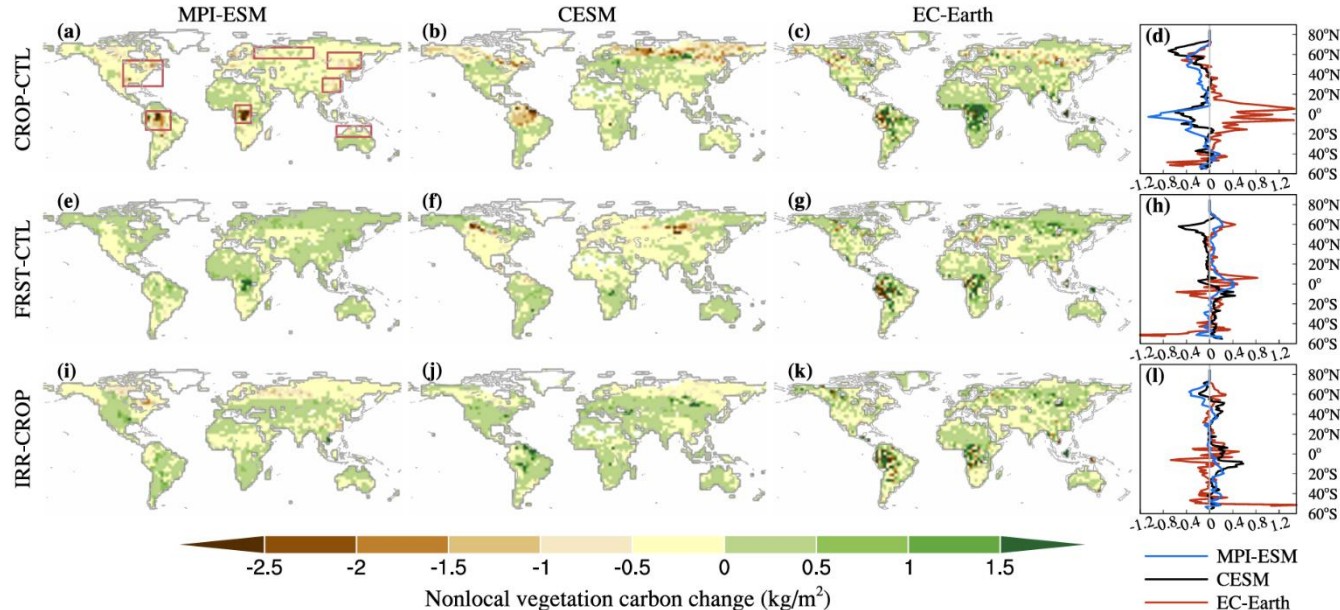

**Figure 3: Nonlocal effects on vegetation carbon of the last 30 years in the 160-year simulation period using MPI-ESM, CESM, and EC-Earth after an idealized change of 50 % of all grid cells (a-c) to cropland expansion, (e-g) to afforestation, and (i-k) to cropland expansion with irrigation. Red boxes in (a) define areas used for the calculation of regional averages in Fig. 5 and 6. Panels d, h, l are latitudinal means over the land areas.**

In the IRR scenario, an increase in nonlocal soil moisture (Fig. C3) consistently induces higher cVeg across most regions and among the three models (Fig. 3i-k). In the low latitudes and mid-latitudes, cVeg is generally simulated to increase, especially over the western Amazon and central Congo Basin rainforests, north-central America, and Eurasia. However, for MPI-ESM and CESM, cVeg in the Northern Hemisphere boreal latitudes (50° N-90° N) is simulated to slightly decrease. In the special case of MPI-ESM, the loss of cVeg is large enough to offset the cVeg increases observed elsewhere globally. Furthermore, a high percentage of bare land cover in the boreal grid cells, with cVeg unaffected by nonlocal BGP effects, reduces the grid average nonlocal BGC effects. Generally, despite some inconsistency between the global integrals among the three models, the spatial distribution of nonlocal cVeg changes shows similar features.

For the FRST scenario, MPI-ESM and CESM present mostly opposite nonlocal cVeg effects compared to the CROP scenario; however, EC-Earth presents similar effects compared to the CROP scenario in the entire western Amazon and central Congo Basin regions. MPI-ESM and CESM present minor cVeg increases in the western Amazon region (Fig. 3e-g). In the central Congo Basin region, MPI-ESM presents a large cVeg increase by +4 GtC. Both, MPI-ESM and EC-Earth, present a similar latitudinal pattern for cVeg increases in the Northern Hemisphere, albeit with differing magnitudes. Conversely, CESM presents a substantial cVeg decrease with -6 GtC in the Northern Hemisphere high latitudes.

The model divergence regarding regional nonlocal BGC effects can be attributed to several factors. First, in the CROP scenario, opposite to MPI-ESM and CESM, cVeg increase in EC-Earth over the tropics is related to the wetting, which is related to the

EC-Earth's strong mesoscale effects and higher spatial resolution (De Hertog et al., 2024). Second, in the IRR scenario, opposite to MPI-ESM and CESM, cVeg increase in EC-Earth over the Northern Hemisphere high latitudes is related to two main reasons: (i) The direct flux of surface water and energy to the atmosphere is limited in EC-Earth due to uncoupled water cycle components between land and atmosphere (see Sect. 2.1). Consequently, irrigation induces only minor local and nonlocal BGP climate changes, resulting in correspondingly minor nonlocal BGC effects. (ii) For MPI-ESM and CESM, the decrease

in cVeg aligns with the cooling observed over Northern Hemisphere high latitudes. This cooling is more pronounced in MPI-ESM and relatively minor in CESM (Fig. C2), which is related to increased cloud cover occurring in both models—substantial in MPI-ESM and moderate in CESM (De Hertog et al., 2023). Third, in the FRST scenario, opposite to MPI-ESM and EC-Earth, cVeg decrease in CESM over the Northern Hemisphere high latitudes is related to a strong soil moisture decrease in these regions (Fig. C3).

**3.2.2 Nonlocal soil carbon stock changes**

Usually, cSoil changes are simulated to be consistent with cVeg when cVeg is large, explicable by the fact that the carbon input to cSoil stems from cVeg. However, respiration by soil heterotrophs is climate-dependent and largely independent of the climate-dependency of the vegetation processes. Overall, an alignment of cVeg and cSoil changes apply to many regions for all three scenarios and three models, particularly the tropics, and occasionally to high latitudes in the Northern Hemisphere.

For the CROP scenario, CESM and EC-Earth show that cSoil changes typically align with cVeg changes in most regions but with smaller magnitudes (Fig. 3b, c and Fig. 4b, c). An exception is EC-Earth in the Northern Hemisphere high latitudes, simulating +1 GtC cSoil gains and -6 GtC cVeg losses. MPI-ESM, however, simulates opposite changes, with cSoil gains of +2 and +0.3 GtC and cVeg losses of -11 and -12 GtC in the Northern Hemisphere high latitudes and low latitudes, respectively (Fig. 3a and Fig. 4a). For EC-Earth, the mechanism involves dynamic shifts of vegetation types driven by climate change; our

simulations show that with this model behavior, a reduction in cVeg and, subsequently, a significant input of cVeg into soil pools occurs. Concurrently, the lower temperature in the Northern Hemisphere high latitudes suppresses the decomposition rate of soil organic matter (Fig. C2c).

In the FRST and IRR scenarios, EC-Earth also simulates cSoil decreases in contrast to cVeg increases. This implies that climate change negatively impacts soil carbon sequestration following these two LCLMC scenarios.

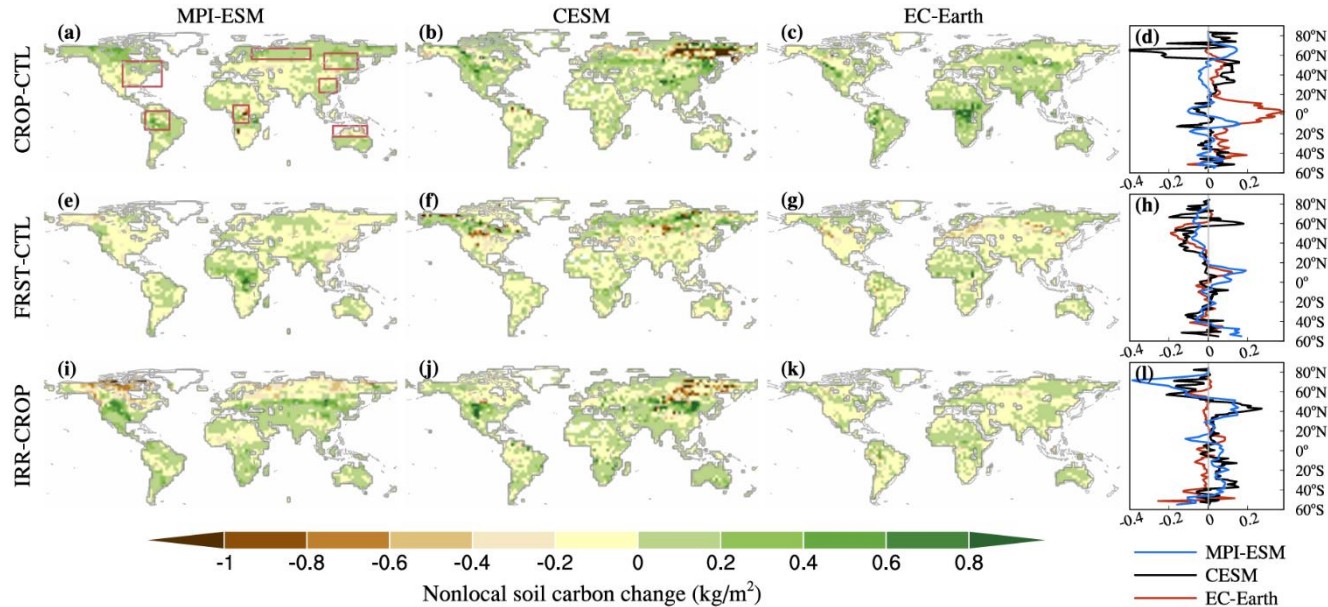

**Figure 4: Nonlocal effects on soil carbon of the last 30 years in the 160-year simulation. See Fig. 3 for details.**

## 3.3 Magnitude of nonlocal to total BGC effect

We aggregate results to a few core regions. These regions were chosen because they exhibit a large absolute nonlocal signal and the signal across models is consistent (Fig. 5, 6). Additionally, we choose regions across various latitudes to capture latitudinal diversity.

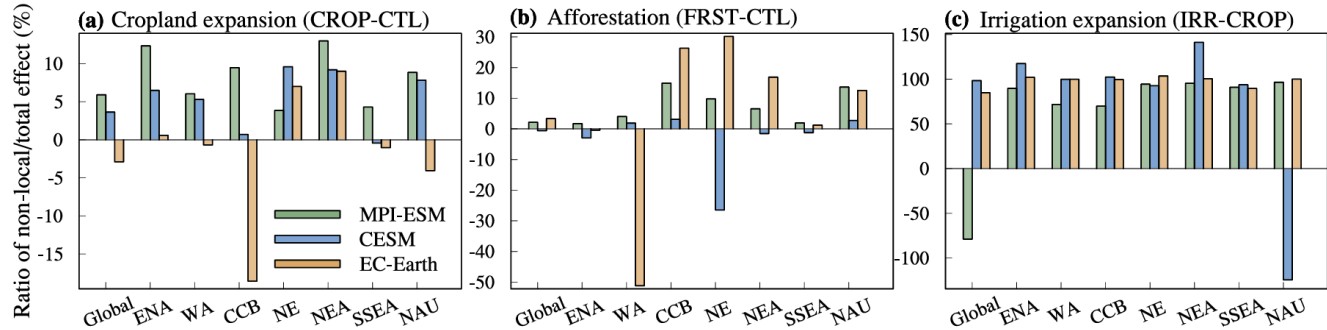

**Figure 5: Relative contribution of the nonlocal to total effect of LCLMC on vegetation carbon of the last 30 years in the 160-year simulation period using MPI-ESM (green), CESM (blue), and EC-Earth (orange) for (a) cropland expansion, (b) afforestation, and (c) irrigation of cropland expansion. Values are separated into the global integral (global) and regional means for eastern North America (ENA), western Amazon (WA), central Congo Basin (CCB), northern Eurasia (NE), northeastern Asia (NEA), southern Southeast Asia (SSEA), and northern Australia (NAU). Regions are defined within the red boxes in Fig. 3a, and only terrestrial areas are considered.**

In the CROP scenario, total cVeg and cSoil effects are negative across all models globally and in selected regions. The globally integrated nonlocal effect on cVeg constitutes approximately 6 %, 4 %, and 3 % of the total effect for MPI-ESM, CESM, and EC-Earth, respectively. This ratio can exceed 12 %, 9 %, and 8 % in regions such as eastern North America, northeastern Asia,

and northern Eurasia, for MPI-ESM, CESM, and EC-Earth, respectively. There is less model consistency in ratios for nonlocal cSoil changes in the CROP scenario. EC-Earth simulates negative signals of the nonlocal to total effect, both globally and in the selected key regions.

In the IRR scenario, we find a pronounced relative global nonlocal cVeg response constituting 98 % and 85 % of the total cVeg gains for CESM and EC-Earth, respectively. In contrast, MPI-ESM shows nonlocal cVeg losses opposite to the total gains, with a nonlocal to total ratio of -79 %. Regionally, CESM shows nonlocal cVeg gains exceeding total gains in eastern North America (117 %) and northeastern Asia (141 %). For cSoil effects, EC-Earth shows globally integrated nonlocal cSoil losses constituting 66 % of total cSoil losses, whereas MPI-ESM exhibits nonlocal cSoil gains opposing total losses, with a ratio of -169 %. The main reason behind the pronounced relative importance of the nonlocal cVeg response is that the land management change of irrigation per se, in the absence of land-cover change, does not induce carbon stock changes directly. Consequently, the local BGC effects are mostly a response to the changes in local climate through irrigation. The local and nonlocal BGC effects are of comparable magnitude.

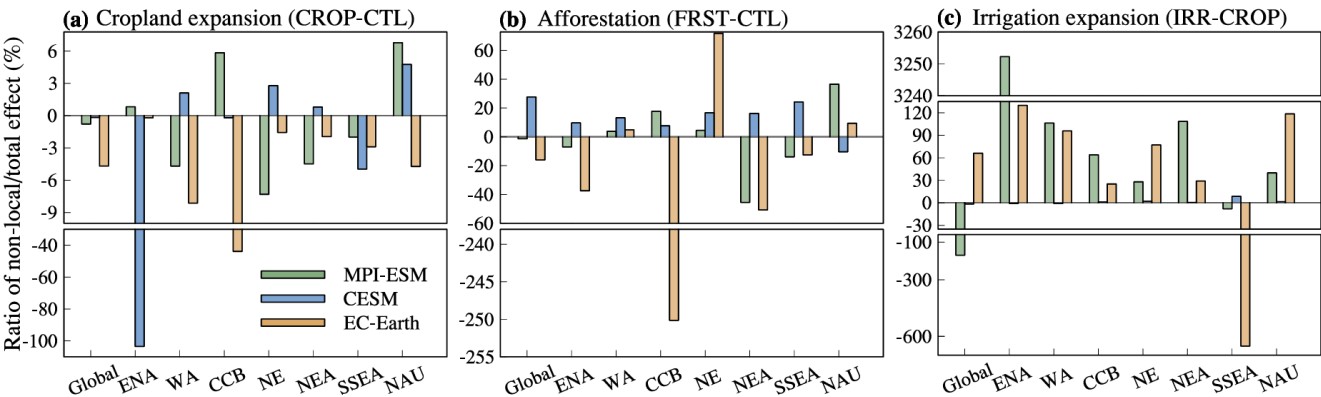

**Figure 6: Relative contribution of the nonlocal to total effect of soil carbon of the last 30 years in the 160-year simulation period. See Fig. 5 for details.**

In the FRST scenario, apart from cSoil changes in CESM and EC-Earth, total BGC effects show a positive trend globally and in specific regions. For CESM, total cSoil losses are observed except in the central Congo Basin and northern Eurasia, while for EC-Earth, total cSoil gains are observed except in northern Eurasia. The relative importance of nonlocal cVeg changes in EC-Earth surpasses 26 %, 30 %, 16 %, and 12 % in central Congo Basin, northern Eurasia, northeastern Asia, and northern Australia, respectively. The values are similar for MPI-ESM with 13 % in central Congo Basin and northern Australia. Notably, nonlocal cSoil changes in EC-Earth constitute over 70 % of total cSoil changes in northern Eurasia while for CESM, nonlocal cSoil changes represent only about 27 % of the total cSoil changes globally, and over 9 % in all key regions except northern Australia. The relative nonlocal effect also exceeds 17 % and 36 % in central Congo Basin and northern Australia, respectively, for MPI-ESM.

The model divergence regarding the nonlocal-to-total ratio derives from the divergence in both nonlocal and total BGC effects. The ratio of cSoil, for instance, is less consistent across models for the CROP scenario. EC-Earth presents negative ratios

across regions, while the sign of the ratio in CESM and MPI-ESM varies by region. Ratio signs of the CROP scenario are related to an overall decrease in total cSoil in the selected regions and globally across all three models, alongside an increase in nonlocal cSoil in these regions for EC-Earth (see Sect. 3.2.2). In the IRR scenario, the global negative ratio of cVeg in MPI-ESM is largely influenced by nonlocal cVeg losses in the Northern Hemisphere boreal regions, driven by strong cooling effects in these regions.

## 3.4 Time of emergence

Generally, nonlocal cVeg changes emerge within less than 40 years (Figs. 7 and E1) for the majority of the hospitable land area for all LCLMC scenarios. ToE shows a similar pattern of variation with latitude across models for all three scenarios (Fig. 7). In the tropics and Northern Hemisphere high latitudes, the ToE occurs earlier, typically within 30 years. In the tropics, this early ToE is dominated by the western Amazon and central Congo Basin, i.e. for rather forested regions; here ToE can be even shorter than ten years, depending on the model and scenario (Fig. E1). The mid-latitudes show a later ToE, with different magnitudes across models. For example, eastern North America typically show a later ToE which is indicated by the relatively flat trend in the temporal development of carbon pools during the initial decades (Figs. E1 and D4). Crop- and grasslands take a considerable fraction of land in eastern North America, indicating that the response of those land cover types is slower than that of forests. However, for MPI-ESM and CESM, the nonlocal BGC effect in eastern North America reaches a magnitude comparable to that in northern Eurasia, northeastern Asia, and southern Southeast Asia by the end of the simulation period (see Fig. 3 and Appendix D for details). This suggests that the nonlocal climate impact on crop- and grasslands persistently accumulates over time, and ultimately becomes comparable to that on forests.

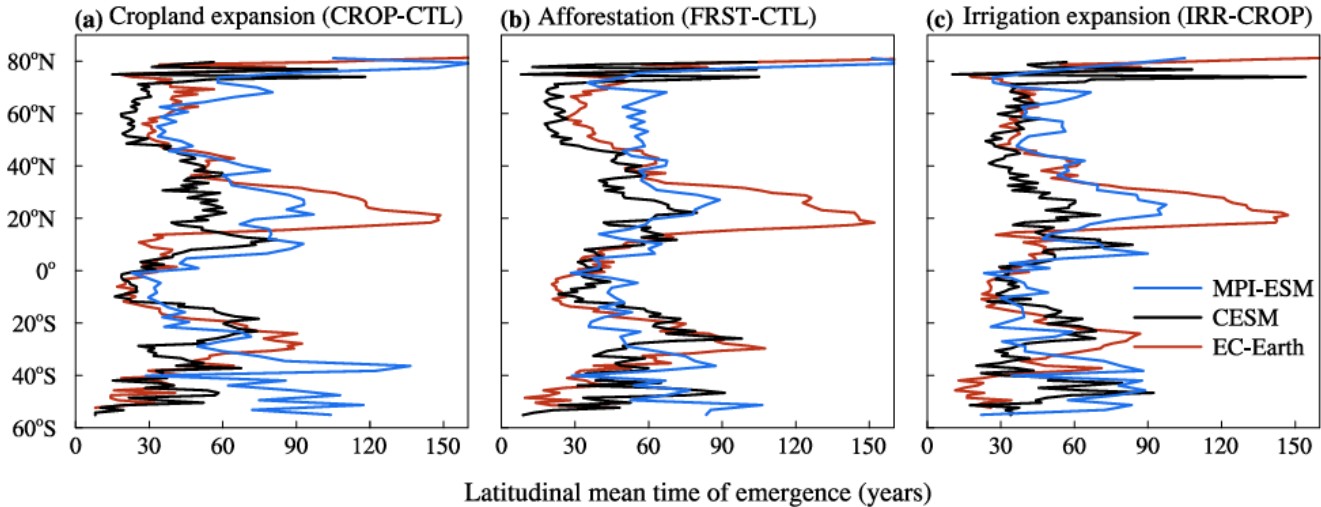

**Figure 7: Latitudinal mean time of emergence for nonlocal vegetation carbon changes surpassing natural variability in the (a) cropland expansion, (b) afforestation, and (c) irrigation of cropland expansion scenario (c). Results are shown for MPI-ESM (blue), CESM (black), and EC-Earth (red).**

For cSoil, the ToE is also generally shorter than 40 years for the majority of the hospitable land area for all scenarios and models (Figs. 8 and E2). The latitudinal mean ToE shows smaller variation for all models and scenarios. The ToE for cSoil is typically shorter than that for cVeg, which is related to the relatively smaller internal variability of cSoil.

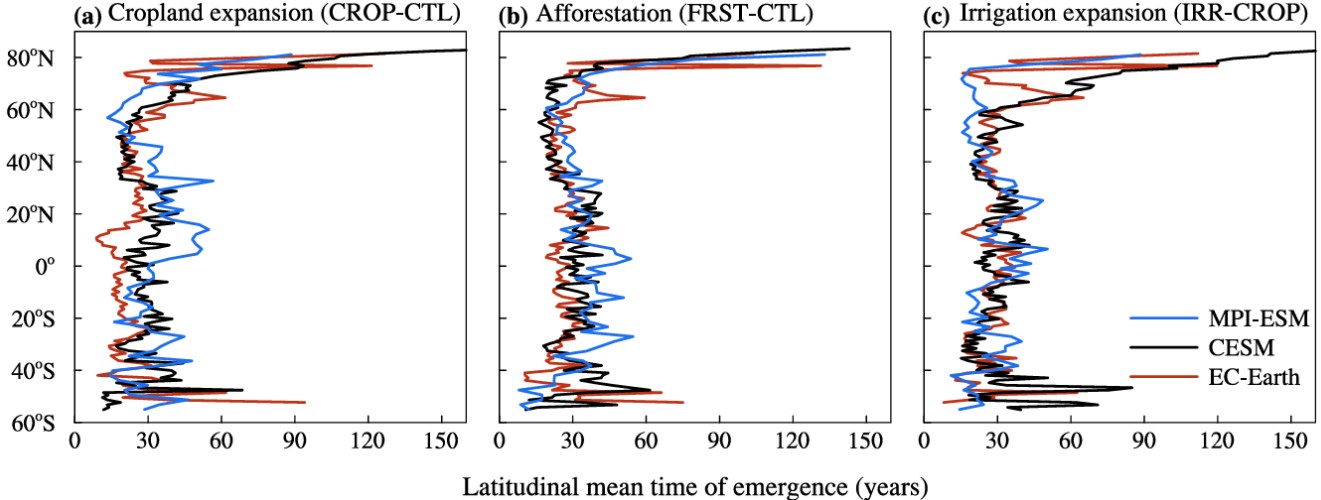

**Figure 8: Latitudinal mean time of emergence for nonlocal soil carbon changes surpassing natural variability. For details see Fig. 7 | Same as Fig. 7 but for soil carbon.**

In the mid-latitudes, the ToE for nonlocal cVeg changes in EC-Earth is generally later than in the other two models across all three scenarios. This delay is related to the impact of dynamic vegetation competition and replacement. Additionally, the ToE is further delayed in the FRST scenario due to the gradual establishment of tree physical properties.

### 3.5 Impacts of temperature and soil moisture on nonlocal BGC effects

Except for EC-Earth's cSoil sensitivity to soil moisture, the sensitivity of cVeg and cSoil to temperature and soil moisture is highly consistent across three models and scenarios in global distribution and sign. We discuss the CROP scenario and present the signals of the FRST and IRR scenario with Fig. D1-D4.

All three models agree that in the low latitudes, elevated nonlocal temperatures and decreased nonlocal soil moisture induce reductions in nonlocal cVeg. The magnitude of cVeg sensitivity to nonlocal BGP effects is particularly high in the western Amazon and central Congo Basin regions (Fig. 9a-c and e-g). Obviously, less soil moisture restricts plant assimilation. Elevated temperatures induce an increase in gross primary productivity and even more in autotrophic respiration, which in the end leads to a decrease in cVeg (Lawrence et al., 2019; Reick et al., 2021; Smith et al., 2014). In the Northern Hemisphere boreal latitudes, increased temperatures positively influence cVeg.

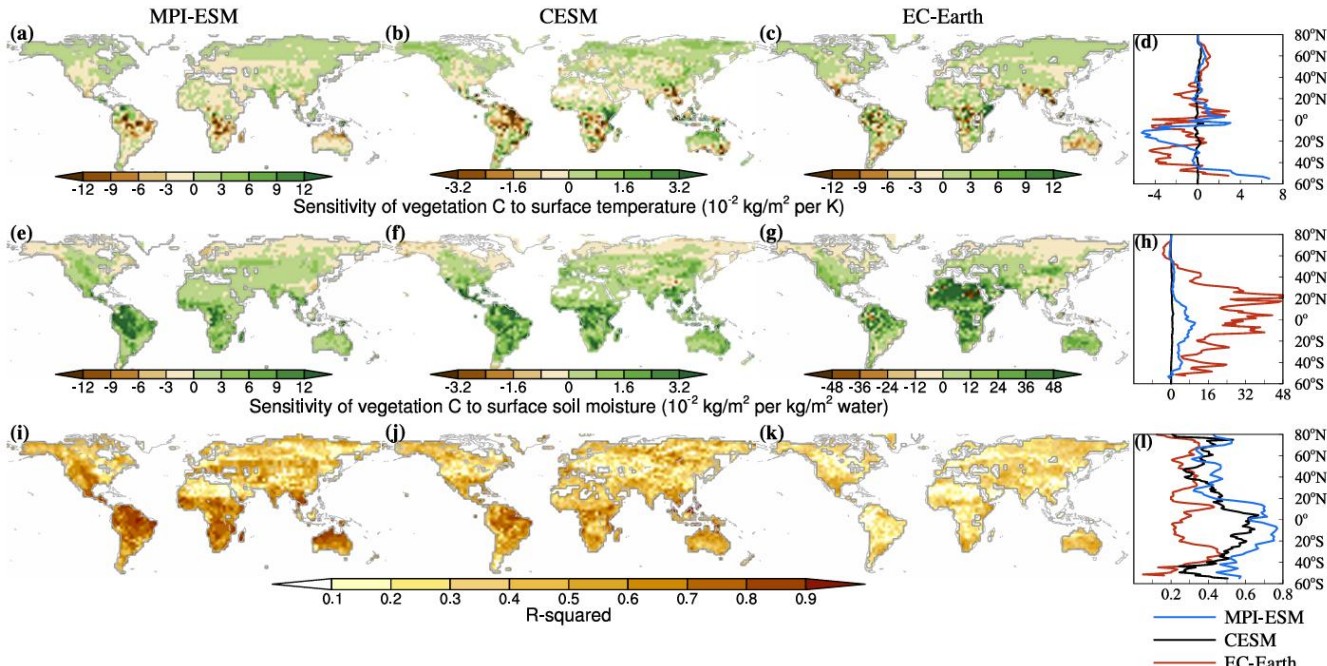

**Figure 9: Attribution of vegetation carbon changes to changes in near-surface air temperature (a-d) and surface soil moisture (e-h) and the respective R$^2$ values (i-l) from a multiple linear regression analysis for the cropland expansion scenario for MPI-ESM, CESM, and EC-Earth (see Fig. D1 and D3 for afforestation and irrigation of cropland scenarios, respectively). Note that the value scale differs between models. Panels d, h, l are latitudinal means over the land areas.**

Generally, while the distribution of cVeg sensitivity is similar across models, the magnitude differs. In low latitudes, the cVeg of CESM decreases by -11 GtC for every Kelvin increase in temperature. This is less than the magnitudes for MPI-ESM and EC-Earth, which are -18 GtC K$^{-1}$ and -19 GtC K$^{-1}$, respectively. In the central Congo Basin and western Amazon regions, the sensitivity difference of cVeg is even greater among the three models. MPI-ESM and EC-Earth simulate a cVeg loss of about -10 GtC K$^{-1}$ more than CESM in the central Congo Basin region, while MPI-ESM experiences a cVeg loss of about -12 GtC K$^{-1}$ more than CESM and EC-Earth in the western Amazon region. For every millimeter increase in soil moisture, cVeg increases the most in EC-Earth and the least in CESM. For example, in the low latitudes, cVeg increases by 85, 231, and 821 GtC for CESM, MPI-ESM, and EC-Earth, respectively (Fig. 9h). For MPI-ESM and CESM, the multiple linear regression model in the low latitudes provides a better explanation of the cVeg changes, with the average coefficient of determination (R$^2$) being 0.70 and 0.57 for MPI-ESM and CESM, respectively (Fig. 9I).

Both, MPI-ESM and CESM, show that in most regions global increases in temperature and soil moisture lead to decreased cSoil due to accelerated decomposition rates. This feature remains consistent across scenarios (Fig. D1-D4), except for EC-Earth, where soil moisture plays an opposing role, with increasing soil moisture correlating with increased cSoil. The magnitude of cSoil sensitivity to nonlocal BGP effects is particularly high in the western Amazon and central Congo Basin regions. Except for a similar pattern, the models present a large difference in the magnitude. For example, in the low latitudes, the cSoil sensitivity to temperature is one order of magnitude smaller for CESM (-8 GtC K$^{-1}$) and EC-Earth (-12 GtC K$^{-1}$) than

for MPI-ESM (-106 GtC K$^{-1}$). For every millimeter increase in soil moisture, cSoil typically gains in EC-Earth and loses in the other two models globally. For example, in the low latitudes, cSoil increases by 20 GtC for EC-Earth and declines by -94 and -9 GtC for MPI-ESM and CESM, respectively. For MPI-ESM, the multiple linear regression model in the low latitudes and Northern Hemisphere high latitudes provides a better explanation of the cSoil changes (Fig. 10l). CESM and EC-Earth present a similar pattern but smaller magnitude. Overall, MPI-ESM has the highest R$^2$ in the low latitudes (0.58) followed by CESM (0.41) and EC-Earth (0.29).

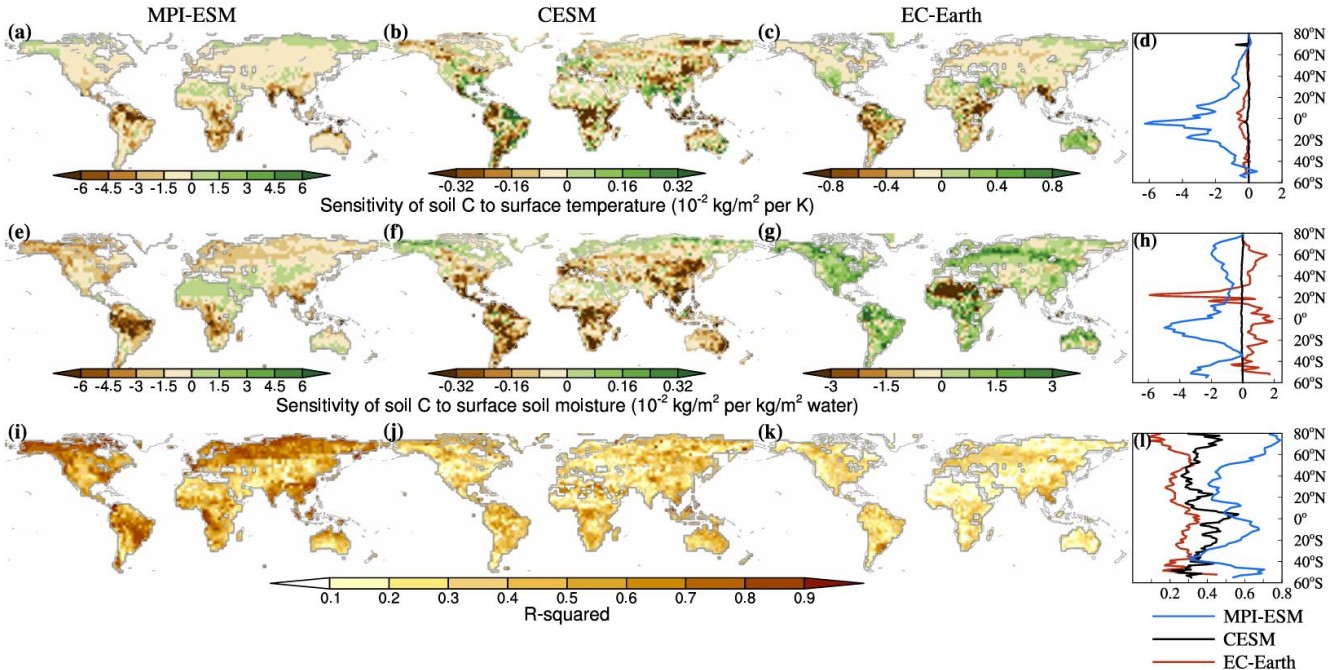

**Figure 10: Attribution of soil carbon changes to changes in near-surface air temperature (a-d) and surface soil moisture (e-h) and the respective R$^2$ values (i-l) for the cropland expansion scenario (see Fig. D2 and D4 for afforestation and irrigation of cropland scenarios, respectively). For details see Fig. 9 | Same as Fig. 9 but for soil carbon.**

The model divergence regarding the sensitivity depends on each ESM and its respective land surface scheme, such as how it represents respiration, photosynthesis, and dynamic vegetation. In our study, a more complex carbon cycle response in EC-Earth is related to the dynamic global changes in vegetation types. In addition to the metabolic responses of plants and soil to nonlocal climate changes, competition and sequential replacement among various plant functional types also influence the carbon cycle. This complexity explains the lower R² value for EC-Earth compared to the other two models in both nonlocal cVeg and cSoil regression models. Additionally, the positive cSoil response to soil moisture in EC-Earth, opposite to the negative responses in the other two models, is related to the fact that nonlocal cSoil changes are driven by nonlocal cVeg changes, which exhibit a strong positive relationship with soil moisture.

## 4. Discussion

### 4.1 Summary and broader relevance

The nonlocal BGC effects accumulate as a result of the persistent nonlocal BGP effects induced by large-scale LCLMCs. The nonlocal changes in cVeg and cSoil appear substantially within the first 40 years for all three scenarios and models. For the CROP scenario, the signals even emerge within the first ten years in the western Amazon and central Congo Basin regions. By the end of our 160-year simulation period, the global nonlocal cLand changes by several to dozens of GtC. The nonlocal BGC effects are often stronger in the western Amazon and central Congo Basin region compared to other regions. For all scenarios, regionally the nonlocal BGC effects are comparable to the total effects, especially for the IRR scenario, in which the nonlocal cVeg and cSoil changes usually approach or even exceed the total effects. The cVeg and cSoil decreases with increasing temperature in the low latitudes, whereas the cVeg increases while the cSoil decreases with increasing soil moisture, except for the simulations with EC-Earth. This major model consistency in sensitivity supports our hypothesis that the model divergence in nonlocal BGC effects is the result of distinct nonlocal climate effects (Fig. C2 and Fig. C3). The carbon cycle sensitivity to temperature and soil moisture is consistent among the three scenarios.

The nonlocal BGC effects are typically more pronounced for the CROP scenario globally and in key regions like the western Amazon and central Congo Basin regions. This holds for all three models. This is due to the more pronounced nonlocal BGP effects of the CROP scenario, as the sensitivity of the carbon pool changes with climate is highly consistent across scenarios and consistent with previous research of the low latitudes (Arora et al., 2013; Hubau et al., 2020; Koch et al., 2021; Sullivan et al., 2020). Nonlocal soil moisture changes are dominant, given the magnitude of temperature and soil moisture changes (Fig. C2 and C3) and the fact that one-millimeter soil moisture changes result in larger carbon stock changes than one-Kelvin temperature changes.

The nonlocal BGC effects show an asymmetric response between the CROP and FRST scenario due to LCLMCs patterns. For instance, given the originally high percentage of forest cover over the western Amazon region, the CROP scenario shows an extensive land cover transition to cropland. The FRST scenario, in contrast, only shows a slight transition to forest. This asymmetry leads to smaller remote changes in the FRST scenario for both, temperature (Fig. C2) and soil moisture (Fig. C3), compared to the CROP scenario, particularly in MPI-ESM. Previous studies using observation-based assessments have shown the difference in land surface properties between newly grown young forest and previously lost older forest (Su et al., 2023; Zhang et al., 2024). A new forest can only have the same influence as a mature forest, on local and nonlocal climate, after a substantial period of development. However, the models in this study differ in representing these processes; only EC-Earth (LPJ-GUESS) simulates the gradual establishment of tree physical properties (Döscher et al., 2022; Smith et al., 2014), explaining the delayed growth trend and typically larger ToE for nonlocal BGC effects in EC-Earth's FRST scenario compared to CESM and MPI-ESM (Fig. 2 and 7).

Apart from nonlocal BGP effects, the local BGP effects are more pronounced for the CROP scenario, especially for EC-Earth (De Hertog et al., 2023). This highlights the importance of stopping cropland expansion, which potentially triggers substantial

nonlocal BGC effects, in contrast to the lagging and smaller nonlocal BGC effects from afforestation. Previous studies have demonstrated the priority of stopping deforestation from multiple perspectives. Regarding carbon stock and biodiversity, after decades of development regenerated forests still fall behind the undisturbed primary forest (Lennox et al., 2018; Smith et al., 2020). Additionally taking the economy and society into account, avoiding deforestation is the most cost-effective LCLMC action to mitigate climate change in the short term (Eriksson, 2020).

The IRR scenario has the largest relative magnitude of nonlocal BGC effects among all LCLMCs scenarios. This is mainly because of the substantial nonlocal BGP effects (Fig. C2) and the comparatively minor local BGC effects compared to the CROP and FRST scenarios. Irrigation has gained attention due to its significant hydrological and climatic impacts (Devanand et al., 2019; Leng et al., 2015; Mahmood et al., 2014; Thiery et al., 2020). For MPI-ESM and CESM, in the low latitudes irrigation mitigates the warming and drying trend following cropland expansion (Fig. C2 and C3), and consequently partially

compensates the cVeg losses in these regions (Fig. 3). The nonlocal BGC effects, as a major contributor to land management emissions, run the risk of being overlooked if we concentrate only on irrigated land and local BGC effects.

Overall, we show that the nonlocal BGC effects are typically strong over dense forest regions, such as the western Amazon and central Congo Basin region, for all three models and scenarios. This is consistent with prior research suggesting that regions with high growth potential, such as forests, are particularly vulnerable (Huxman et al., 2004; Knapp & Smith, 2001).

In addition, dense forests experience an earlier ToE than other types. One reason is the higher sensitivity of carbon pools in these regions to nonlocal BGP effects (Fig. 9 and 10; see Sect. 2.4 for the calculation of ToE), which is related to the high biomass density of the forest. While a higher sensitivity weakens the signal-to-noise ratio, the dominating effect is the larger response of forests to the BGP climate changes. For the CROP scenario, the transition from forest to cropland in the western Amazon and central Congo Basin region (Fig. C1) causes substantial nonlocal BGP effects on nearby regions. This is in line

with previous studies that indicate nonlocal BGP effects to be stronger over regions close to LCLMCs compared to more remote regions (Boysen et al., 2020; Butt et al., 2023; Cohn et al., 2019; Crompton et al., 2021). Our findings warn us of the potential risks that come with LCLMCs around old, dense forests.

The nonlocal BGC effects are currently neglected in scientific assessments and political decision-making around land-use change, adaptation and climate mitigation. Our study highlights the importance of considering these effects. A further

consideration is whether nonlocal BGC effects should enter the definition of land-use emissions. The nonlocal BGC effects fall under indirect effects on managed and unmanaged land, accounted for as anthropogenic removals or emissions by the National Greenhouse Gas Inventories (NGHGIs) under UNFCCC rules (Grassi et al., 2018). The indirect human-induced effects represent land carbon pool changes resulting from climate change, atmospheric $CO_2$, nitrogen deposition, and natural disturbances. These changes partly result from LCLMCs; the contribution could be substantial with extensive LCLMCs.

Though not fundamentally different than for other types of human-induced environmental changes. The presented LCLMC-induced climate effects and its result on remote C stock changes highlight that land use, land use-change, and forestry (LULUCF) activities in one country influences the ecosystem fluxes and thus land-use emissions, as defined by the country reporting under UNFCCC, in another country. By contrast, the indirect human-induced effects, including the nonlocal BGC

effects, are categorized as natural, not anthropogenic, land sinks/sources in the global carbon budgets (Friedlingstein et al.,

2023) and in the IPCC Sixth Assessment Report (Canadell et al., 2023). For the NGHGIs, the nonlocal BGC effects on managed land are accounted for, while those on unmanaged land are currently unaccounted for, as NGHGIs typically measure land use emissions on managed land.

To achieve the Paris Agreement's goal of limiting global warming to below 1.5 °C above pre-industrial levels, which necessitates net-zero $CO_2$ emissions around 2050 and subsequent net-zero emissions for all other greenhouse gases in the

second half of the 21st century (Riahi et al., 2022), carbon dioxide removal and negative $CO_2$ emissions are inevitable. The land sector is expected to contribute significantly to this goal, with LCLMCs playing a pivotal role (Humpenöder et al., 2022; Roe et al., 2019). Given that the nonlocal BGC effect is a non-negligible component of LCLMCs emissions, it should be taken into account for consistent budgeting of greenhouse gas fluxes in line with intended climate policies.

## 4.2 Robustness of results

Despite substantial discrepancies in the global integral of nonlocal BGC effects due to regional magnitude differences, the spatial patterns and signs are consistent among models. An exception is the cVeg changes of EC-Earth in the CROP and the FRST scenarios where the signs are different to the other models. This consistency indicates the robustness of the nonlocal BGC effect, while the multi-model approach provides an assessment of model uncertainty.

The model discrepancies stem from two sources: divergence in nonlocal BGP effects and divergence in the carbon cycle

sensitivity to climate change. The nonlocal BGP effects diverge in magnitude and even sign (Fig. C2 and C3). The difference in temperature and soil moisture could reach several degrees Kelvin and millimeter, respectively, in some regions. The divergence of nonlocal BGP effects is related to the divergence in implemented LCLMCs among models. Typically, in EC-Earth, the land cover does not fully change to a target type due to its dynamic global vegetation model. All three models have substantially different irrigation amounts and spatial distributions. Notably, MPI-ESM shows high irrigation amounts in the

boreal latitudes, differing from the other two models, which could explain the substantial cooling there. Except for EC-Earth, the sensitivity patterns and signs are consistent among models, but there is a substantial discrepancy in the magnitude. The sensitivity depends on each ESM and their respective land surface scheme, for example how it represents respiration, photosynthesis, and dynamic vegetation. In our research, EC-Earth is the only model that simulates dynamic changes in the global distribution of vegetation types. The carbon cycle response is therefore more intricate than in the other two models. For

instance, unfavourable climatic conditions (such as warming and drying) usually result in smaller carbon losses than for the other ESMs or even carbon increases in EC-Earth (Fig. 3c, g, k). Although carbon sequestration benefits from plant acclimation to nonlocal BGP effects, the influence of competition and the sequential replacement between various plant functional types depends on the time scale. It could increase cVeg in the long term while decreasing cVeg in the short term, with a portion of substantial dead vegetation carbon transferred to the litter and soil carbon pool. This explains the opposite cVeg and cSoil

changes of EC-Earth for the CROP scenario in the Northern Hemisphere high latitudes, contributing to model divergence. The model divergence in nonlocal BGC effects is the combined results of both nonlocal BGP effects and the carbon cycle sensitivity.

For example, EC-Earth simulates an increase in soil moisture in the low latitudes, for the CROP scenario, opposite in sign and one order of magnitude smaller in magnitude compared with the changes in CESM and MPI-ESM. Nevertheless, this increment plays a key role in the arid tropics, given that cVeg's sensitivity to soil moisture is far greater for EC-Earth than it is for the

other two models. The cVeg ends up with a major increase which is opposite with the cVeg loss in other two scenarios.

The nonlocal BGC effects especially depend on the background climate and $CO_2$ concentration. BGP effects depend on the background climate (Pitman et al., 2011; Winckler et al., 2017b), and the sensitivity of the carbon cycle to climate change is also influenced by the $CO_2$ concentration. In this study, we investigate effects under present-day environmental conditions, which are of greatest relevance to near-term decisions on how to use our land. However, the results may differ under future or

historical conditions.

The nonlocal BGC effects are substantially dependent on the pattern and magnitude of global LCLMCs. In this study, we implement idealized LCLMCs scenarios (see Sect. 2.1). However, some of our findings apply to realistic LCLMCs; for instance, with a similar initial climate, the carbon cycle sensitivity to climate change is highly consistent among scenarios. Apart from that, the adjacent extensive LCLMCs could generate nonlocal BGC effects comparable to our findings in the target

region, considering the LCLMCs typically generate more substantial nonlocal BGP effects nearby (Guo et al., 2024). Our results could serve as an approximate estimation. However, more realistic simulations or emulator development efforts (Nath et al., 2023) are necessary for accurate estimation in application.

**5. Conclusion**

The nonlocal BGC effects accumulate as a result of the persistent nonlocal BGP effects brought on by large-scale LCLMCs.

They affect regions remote from the locations of LCLMCs as unintended, though potentially large effects. The nonlocal BGC effects typically appear within the first 40 years and even emerge within the first 10 years in the western Amazon and central Congo Basin regions under the CROP scenario. By the end of the 160-year simulation period, the global cLand changes by several to dozens of GtC. For the IRR scenario, the nonlocal BGC effects are typically comparable or exceed the total effects with the nonlocal to total ratio for vegetation carbon pools commonly reaching around 90%. Nonlocal BGC effects can be

attributed to nonlocal climate changes such as changes in temperature and soil moisture, with tropical regions being particularly sensitive. In these regions, every Kelvin increase in temperature results in a decrease of over 10 GtC in cVeg. The cVeg response to soil moisture changes varies across models, with each millimeter increase in soil moisture leading to a rise in cVeg of +85 to more than +200 GtC. The priority of stopping cropland expansion is underscored by the fact that the slow regrowth of a new forest induces lagging nonlocal BGC effects in contrast to the quick effects of mature forest loss. For all scenarios,

the signals are often stronger in the western Amazon and central Congo Basin regions. The LCLMCs around old, dense forests run a risk of triggering amplified nonlocal BGC effects in these forest regions due to high biomass density and near source induced nonlocal BGP effects intensification. Though the nonlocal BGC effects are currently neglected in scientific and political assessments, our study highlights their importance. It is essential to reconsider the definition of land-use emissions

and include the nonlocal BGC effects of LCLMCs. This becomes more relevant when LCLMCs are expected to play a pivotal role in achieving the Paris Agreement's goal of limiting global warming below 1.5 °C above pre-industrial levels.

**Appendix A: Distribution of PFTs within crop and forest categories**

PFTs are used in Earth system models to represent the diversity of land cover types within a grid box. These PFTs have specific biochemical and biophysical properties, which are represented by model-specific parameters. The number and specific types of PFTs within the broader crop and forest categories vary among Earth system models: MPI-ESM includes four forest types and one crop type. CESM includes eight forest types and nine crop types, with each crop type having a corresponding irrigated version for irrigation implementation. EC-Earth does not have specific forest types; instead, it uses a natural type that includes coexistence between grass, shrub, and tree PFTs, and it includes four crop types along with corresponding irrigated versions. The distribution of the specific crop or forest PFTs within the respective cropland expansion or afforestation scenario remains constant in the changed grid cells for each ESM (i.e., we did not change the relative importance of, e.g., broadleaf to needleleaf forest types); we only scaled each crop or forest PFTs in such a way that their sum covered the entire hospitable land. For grid cells without any crop or forest PFTs in the year 2014 land cover data set, we calculated a mean latitudinal value of the distribution of specific crop or forest PFTs. We then assumed this as an initial distribution and applied the same scaling as described before to replace all other vegetation of that grid cell.

**Appendix B: LCLMCs implementation in the different ESMs**

The exact implementation depended on the specific way each ESM and their respective land surface scheme handles LCLMCs: For CESM with its land surface scheme CLM5 (Lawrence et al., 2019), we applied the land cover change scenarios using prescribed states of land cover for each year. For MPI-ESM with its land surface scheme JSBACH3 (Reick et al., 2021), we prescribed the transition between land-cover types, thereby also considering effects from gross land-cover changes within a grid cell. EC-Earth uses the 2nd generation dynamic global vegetation model LPJ-GUESS (Smith et al., 2014) which simulates age-structured dynamics of woody vegetation due to plant growth and competition for light, space, and soil resources with a herbaceous understorey. EC-Earth separates between six stand types (natural, pasture, urban, crop, irrigated crop, and peatland). It does not include the option to simulate prescribed forest PFTs, so we could only prescribe the entire natural stand instead of explicit forest for the FRST scenario in EC-Earth. In the natural stand type, ten woody and two herbaceous PFTs are in competition. As a result, depending on the climate, grassland coexists with the forests and shrubs. Additionally, the dynamic vegetation model determines that the physical properties of trees gradually establish depending on biomass buildup through vegetation growth, unlike the immediate physical forest representation in MPI-ESM and CESM after afforestation (Döscher et al., 2022; Smith et al., 2014).

Regarding irrigation implementation, for the MPI-ESM, we adapted and implemented a simple irrigation scheme into JSBACH. It assures water mass conservation in a coupled atmosphere/ocean climate model and maximizes the effect of irrigation to recycle locally available water to the atmosphere by evapotranspiration. Surface runoff and drainage are first collected in a storage reservoir with 20 cm capacity before being transferred to the skin reservoir, filling it completely as long as water is available in the storage reservoir. In contrast to MPI-ESM, CESM and EC-Earth do not have a constraint on water availability. CESM applies daily irrigation to the surface to retain a target soil moisture, while EC-Earth applies daily irrigation to the top of the soil column depending on the water deficit. All three models implement the flood irrigation method. In CESM, irrigation water is first taken from river storage, with additional water drawn from the ocean when river water is insufficient. In EC-Earth, irrigation water is added to the vegetation model LPJ-GUESS without considering its source. This approach does not directly affect EC-Earth's water cycle, as irrigation in LPJ-GUESS only influences land-atmosphere water fluxes in EC-Earth through its impacts on vegetation type and LAI (Döscher et al., 2022).

**Appendix C: Implementation of LCLMCs and resulting remote climate changes**

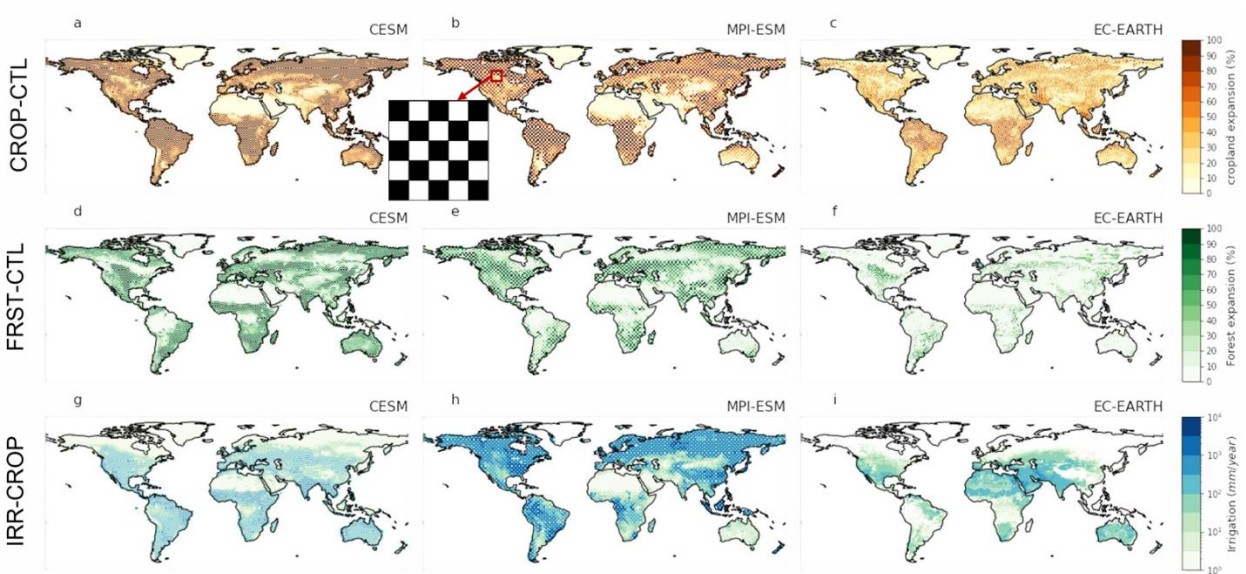

**Figure C1: Land-cover and land management changes implemented in the sensitivity simulations. The cover fraction increase of cropland in the CROP scenario compared to the CTL scenario is shown for CESM (a), MPI-ESM (b), and EC-Earth (c). The cover fraction increase of forest in the FRST scenario compared to the CTL scenario is shown for CESM (d), MPI-ESM (e), and EC-Earth (f). The amount of irrigation implemented in the IRR scenario compared to the CROP scenario is shown for CESM (g), MPI-ESM (h), and EC-Earth (i). Source: De Hertog et al. (2023).**

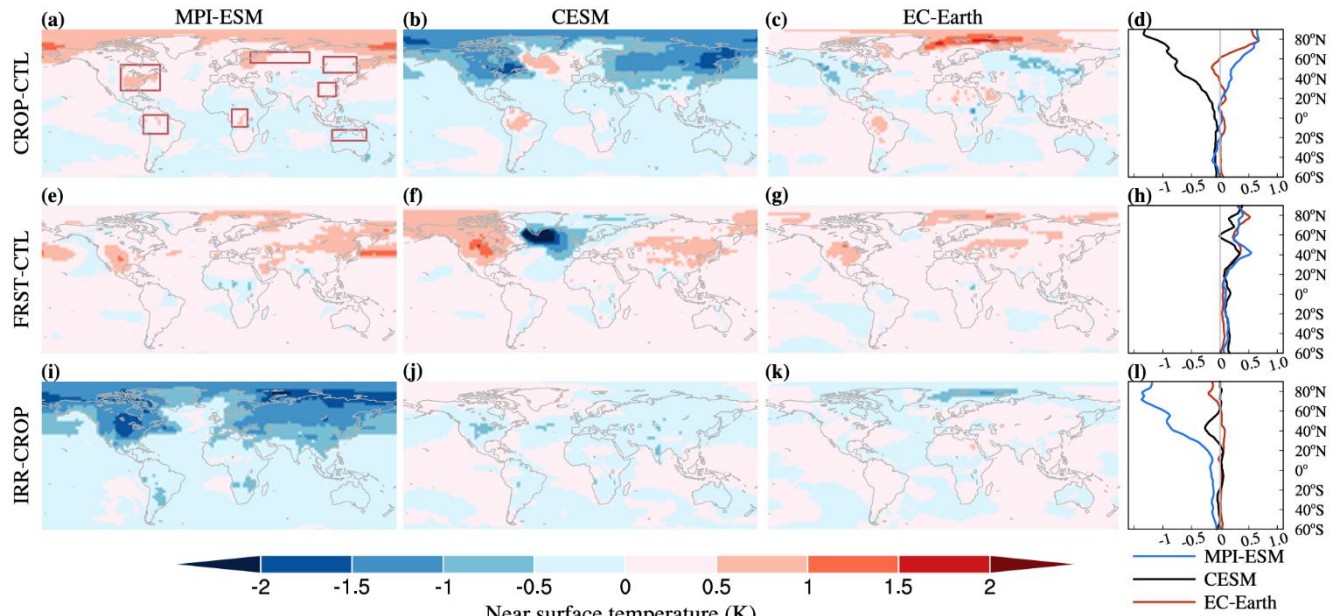

Figure C2: Nonlocal BGP effects on annual mean near-surface air temperature of the last 150 years in the 160-year simulation period using MPI-ESM, CESM, and EC-Earth after an idealized change of 50 % of all grid cells (a-c) to cropland expansion, (e-g) to afforestation, and (i-k) to cropland expansion with irrigation. Panels d, h, l are latitudinal means over the land areas.

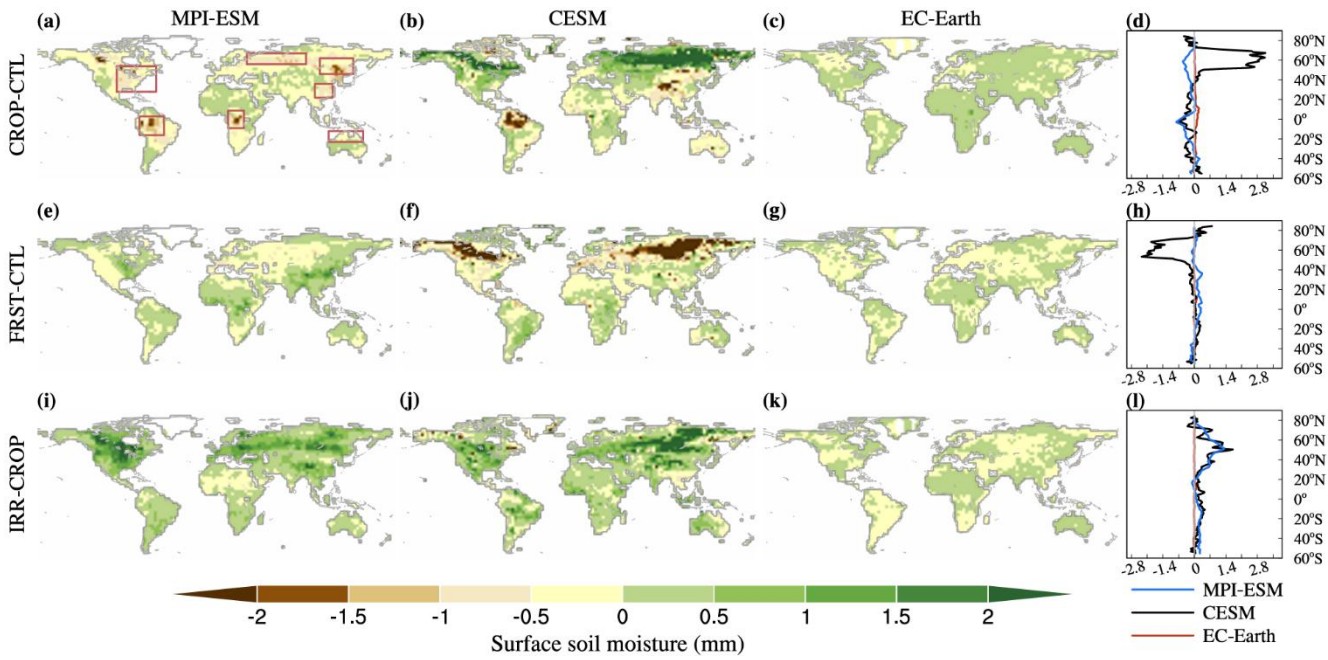

Figure C3: Nonlocal BGP effects on annual mean surface soil moisture of the last 150 years in the 160-year simulation period using MPI-ESM, CESM, and EC-Earth after an idealized change of 50 % of all grid cells (a-c) to cropland expansion, (e-g) to afforestation, and (i-k) to cropland expansion with irrigation. Panels d, h, l are latitudinal means over the land areas.

**Appendix D: Temporal development of regionally integrated nonlocal BGC effects**

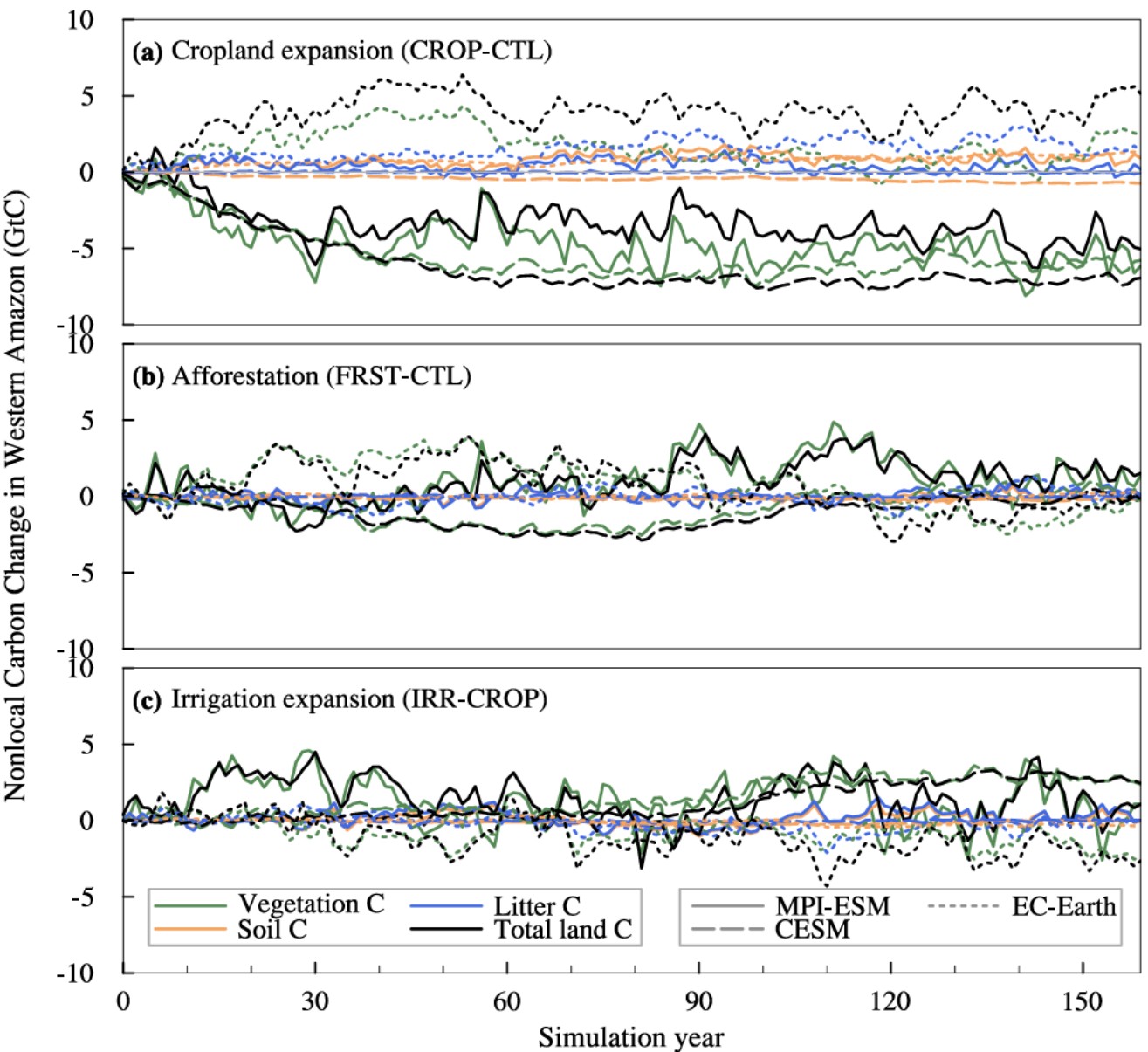

**Figure D1: Simulated nonlocal effect on the development of terrestrial carbon pools in the western Amazon after an idealized change of 50 % of all grid cells (a) to cropland expansion (b) afforestation (c) irrigation of cropland expansion. Carbon pools are separated into vegetation (green), soil (orange), litter (blue), and land as the total terrestrial C pools (black) between results of MPI-ESM (solid lines), CESM (dashed lines), and EC-Earth (dotted lines).**

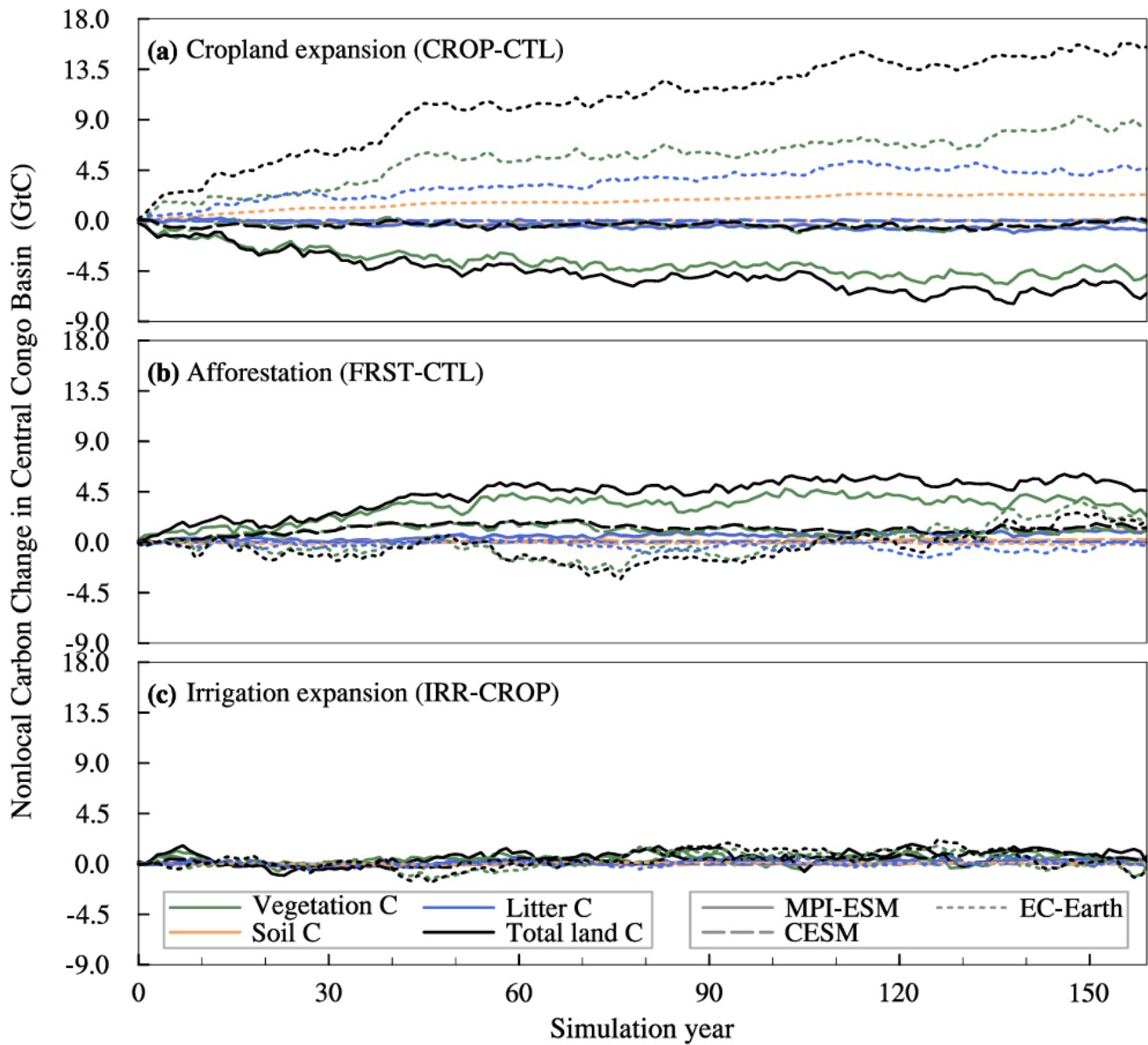

**Figure D2: Simulated nonlocal effect on the development of terrestrial carbon pools in the central Congo Basin. See Fig. D1 for details.**

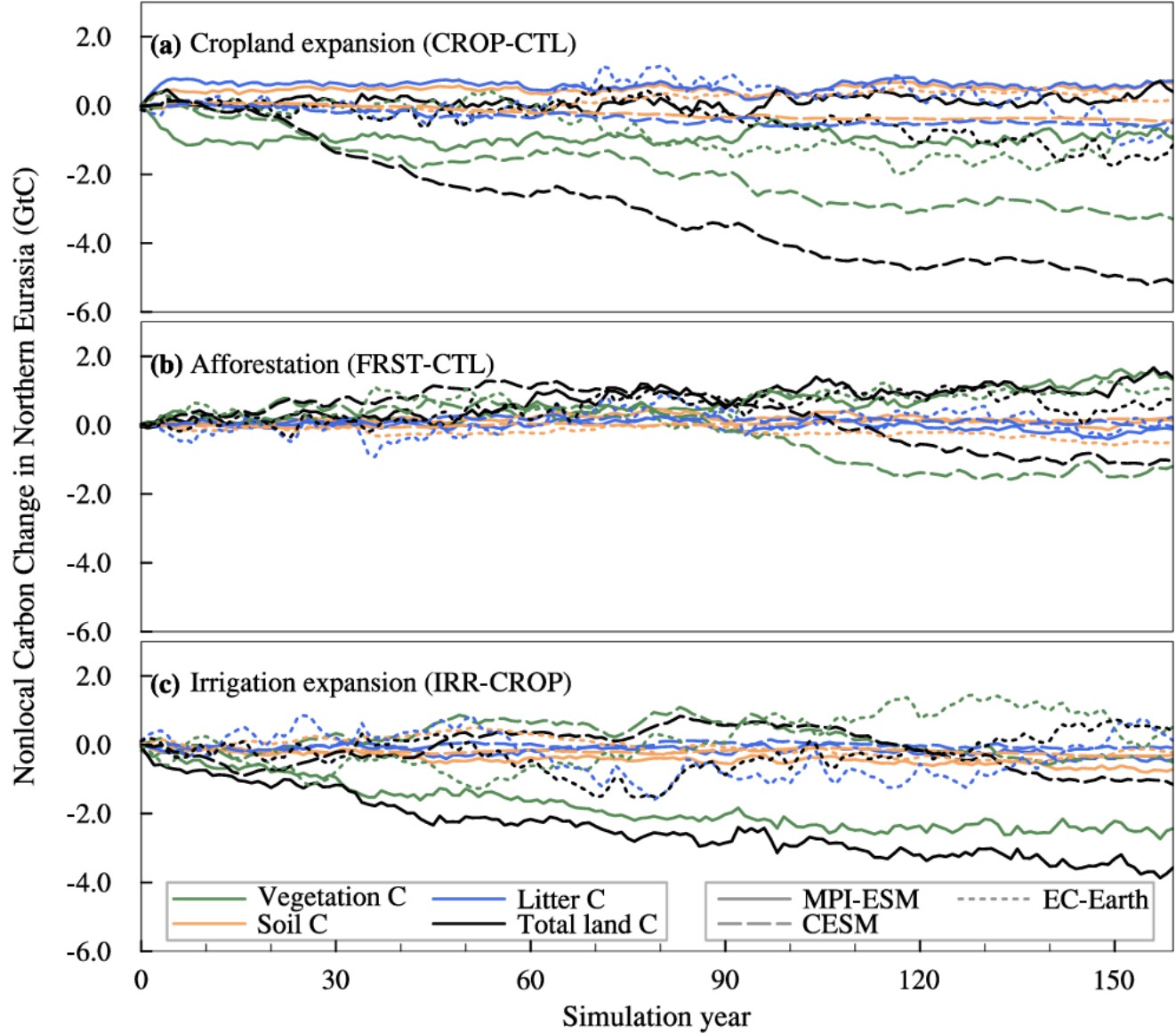

**Figure D3: Simulated nonlocal effect on the development of terrestrial carbon pools in the northern Eurasia. See Fig. D1 for details.**

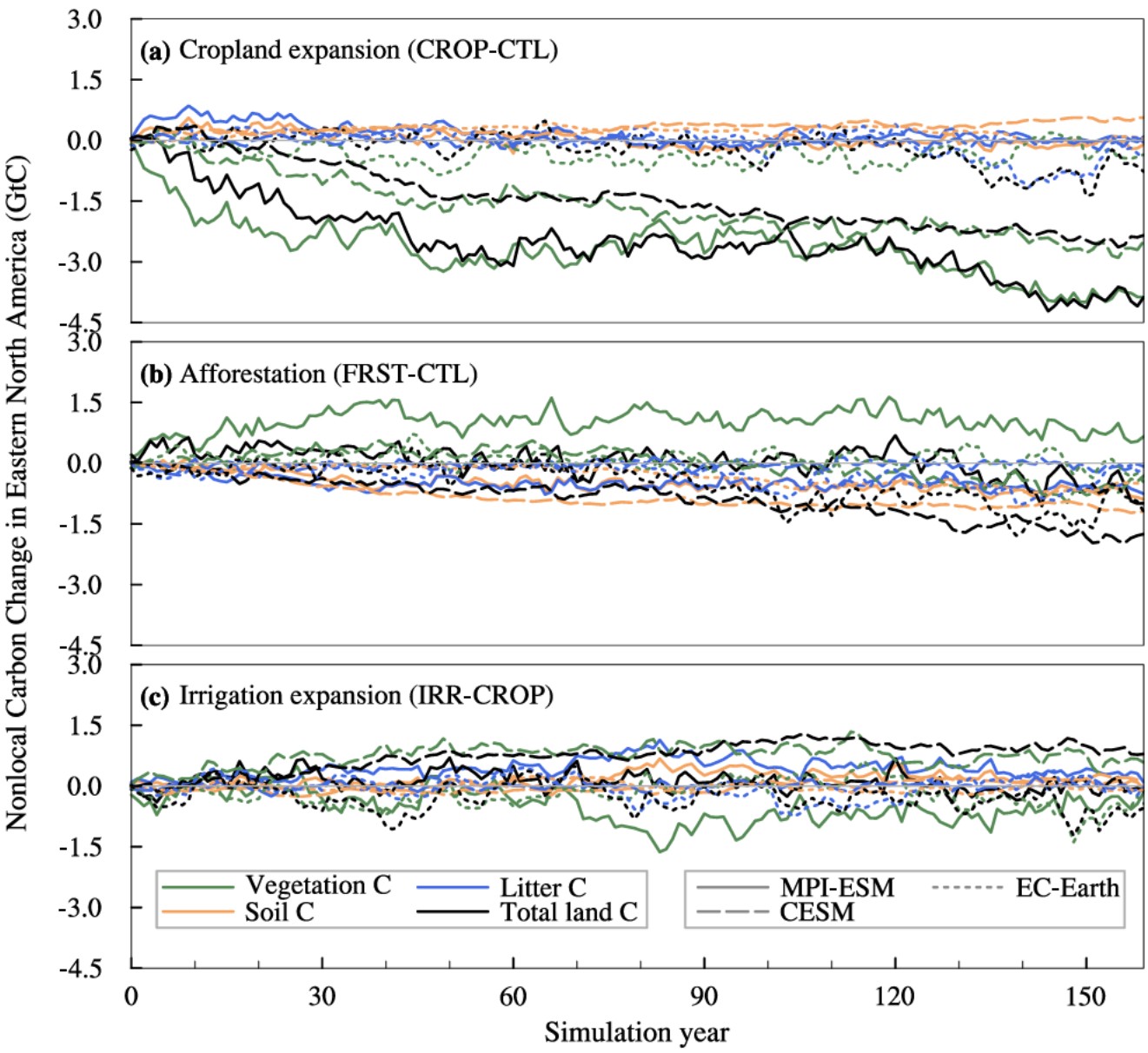

**Figure D4: Simulated nonlocal effect on the development of terrestrial carbon pools in the eastern North America. See Fig. D1 for details.**

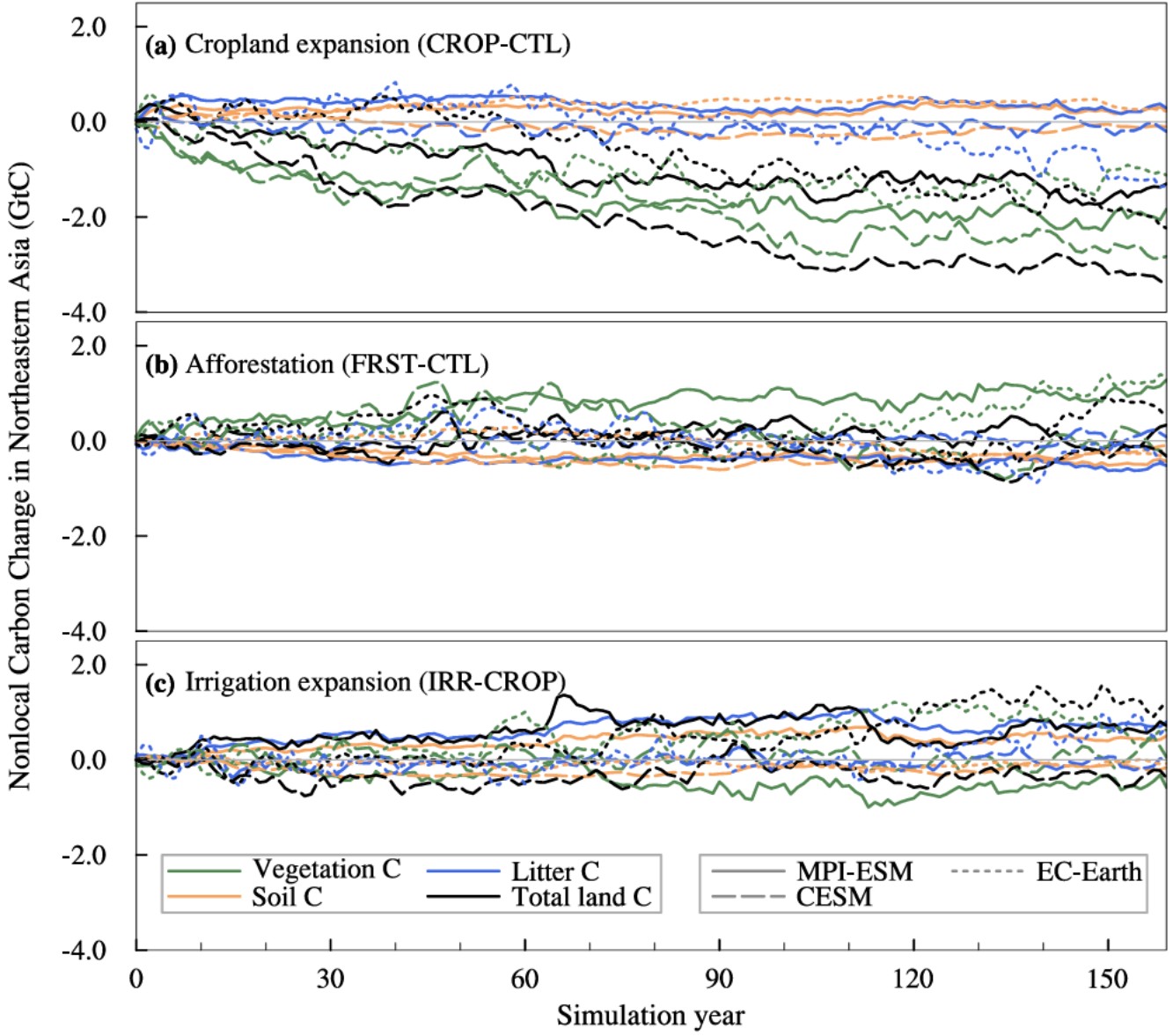

**Figure D5:** Simulated nonlocal effect on the development of terrestrial carbon pools in the Northeastern Asia. See Fig. D1 for details.

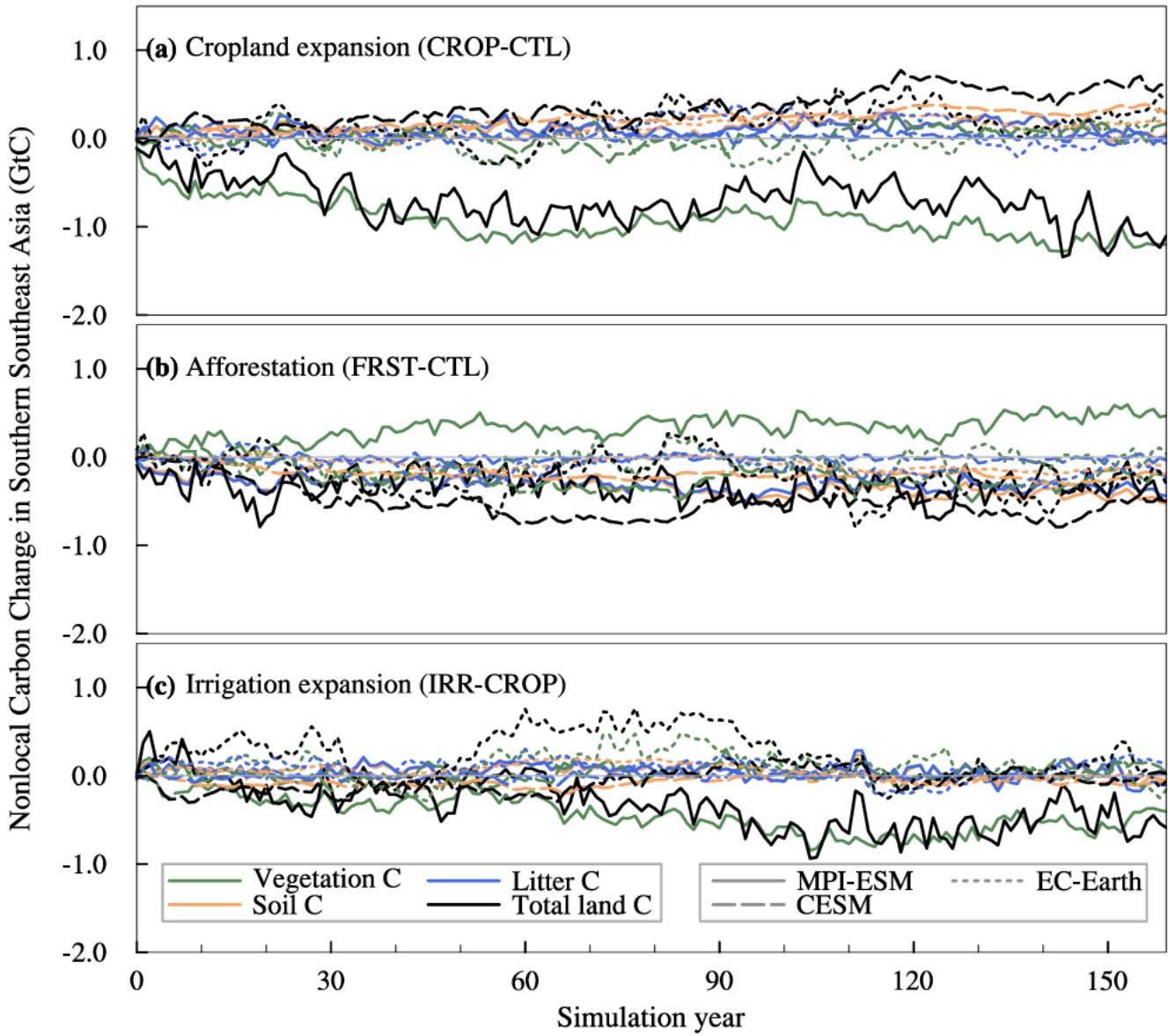

**Figure D6: Simulated nonlocal effect on the development of terrestrial carbon pools in the southern Southeast Asia. See Fig. D1 for details.**

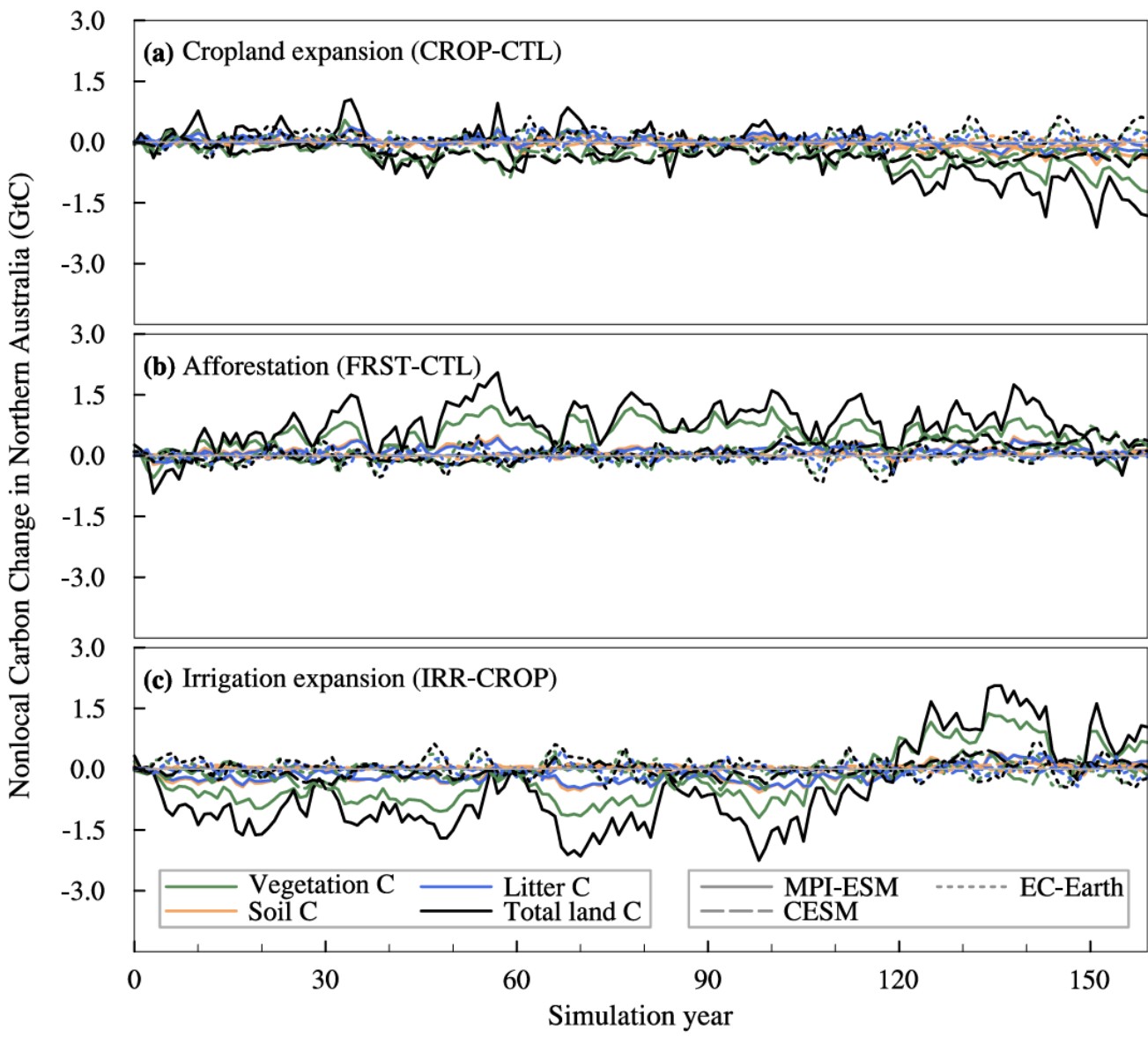

**Figure D7: Simulated nonlocal effect on the development of terrestrial carbon pools in the northern Australia. See Fig. D1 for details.**

## Appendix E: Spatial distribution of ToE

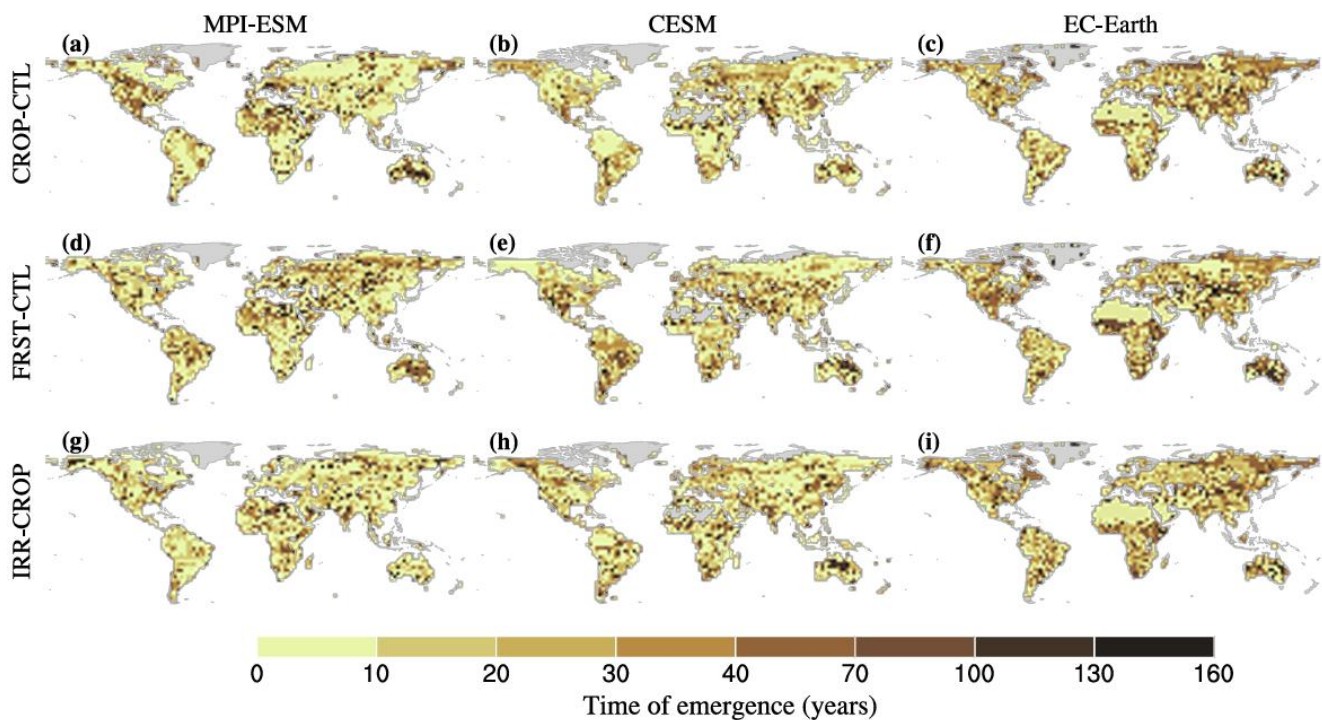

**Figure E1: Time of emergence for nonlocal vegetation carbon changes surpassing natural variability in the cropland expansion (a-c), afforestation (d-f), and irrigation of cropland expansion scenario (g-i). Results are shown for MPI-ESM, CESM, and EC-Earth.**

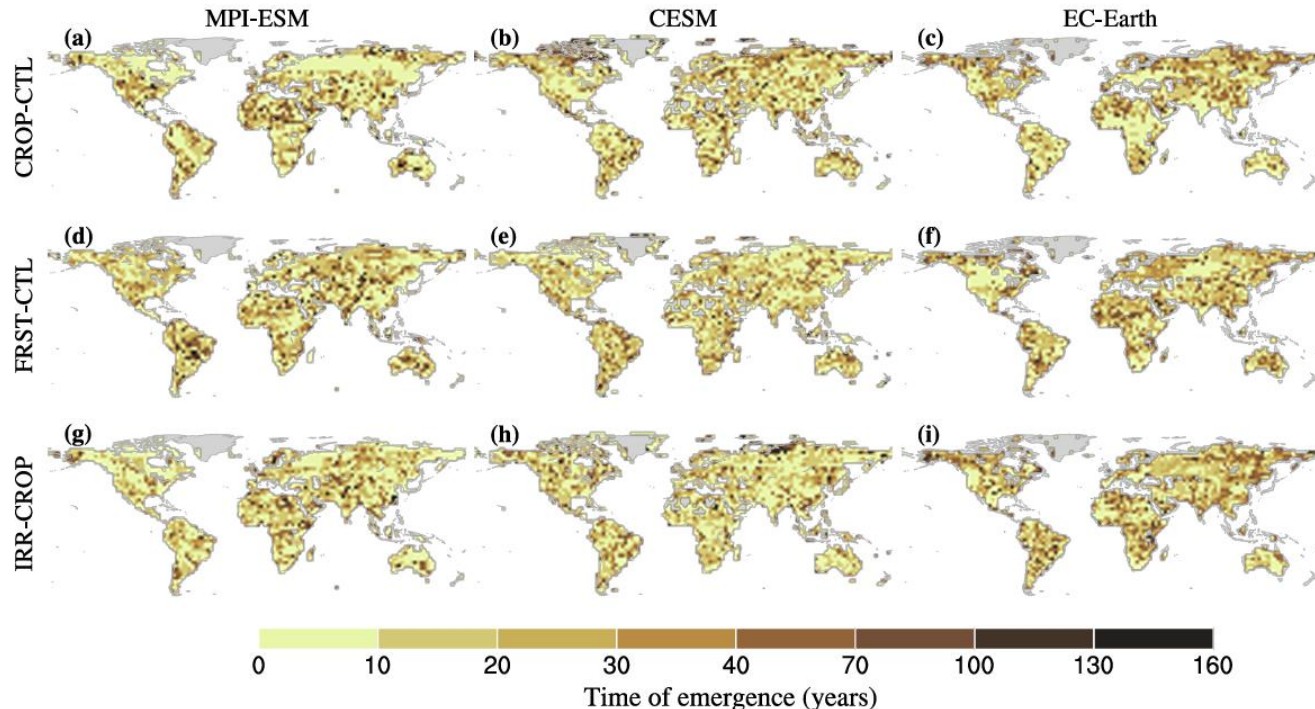

**Figure E2: Time of emergence for nonlocal soil carbon changes surpassing natural variability. For details see Fig. E1 | Same as Fig. E1 but for soil carbon.**

**Appendix F: Impacts of temperature and soil moisture on nonlocal BGC effects for the FRST and IRR scenarios**

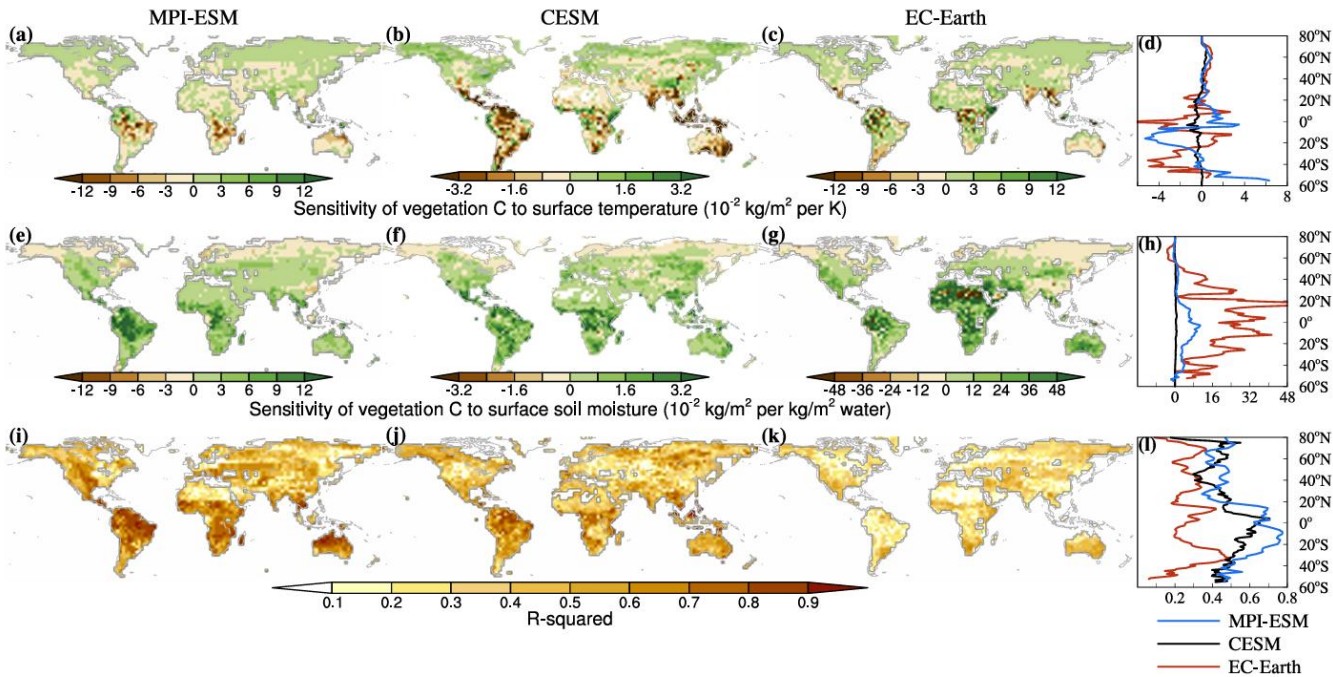

**Figure F1: Attribution of vegetation carbon changes to changes in near-surface air temperature (a-d) and surface soil moisture (e-h) and the respective $R^2$ values (i-l) for the afforestation scenario (see Fig. 9 and F3 for cropland expansion and irrigation of cropland scenarios, respectively). For details see Fig. 9.**

740

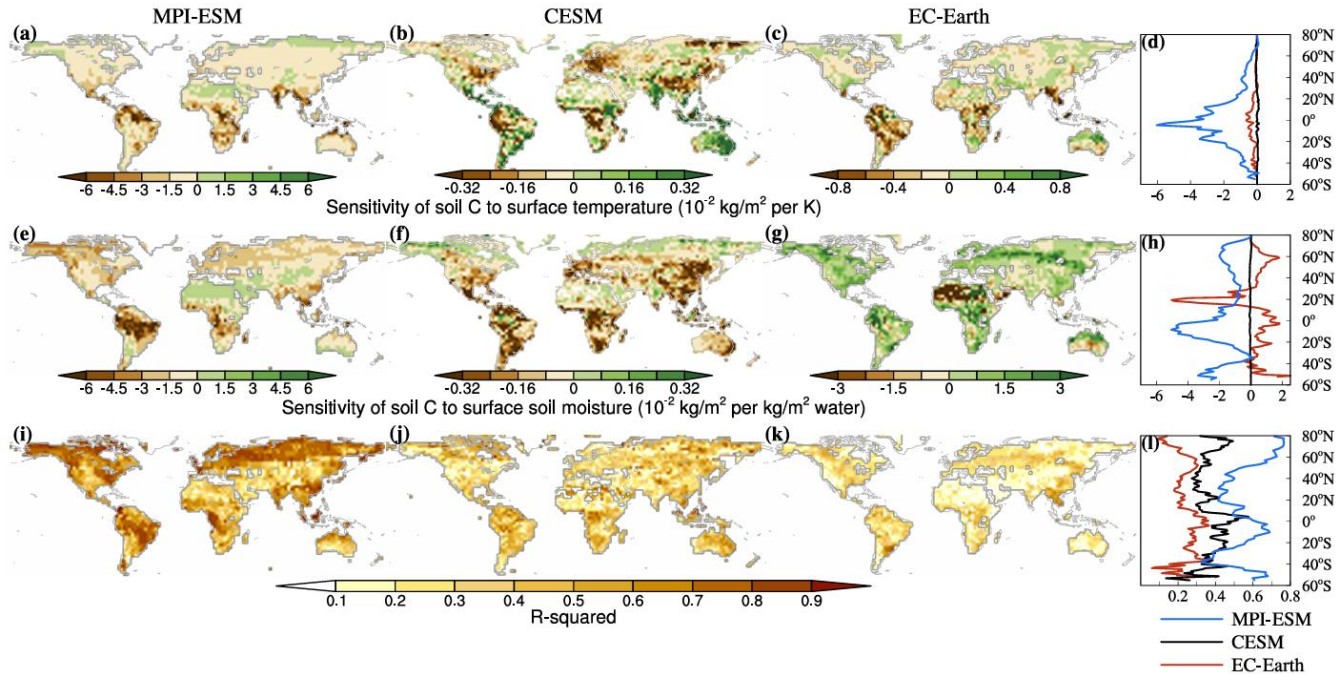

**Figure F2: Attribution of soil carbon changes to changes in near-surface air temperature (a-d) and surface soil moisture (e-h) and the respective R² values (i-l) for the afforestation scenario (see Fig. 10 and F4 for cropland expansion and irrigation of cropland scenarios, respectively). For details see Fig. 9.**

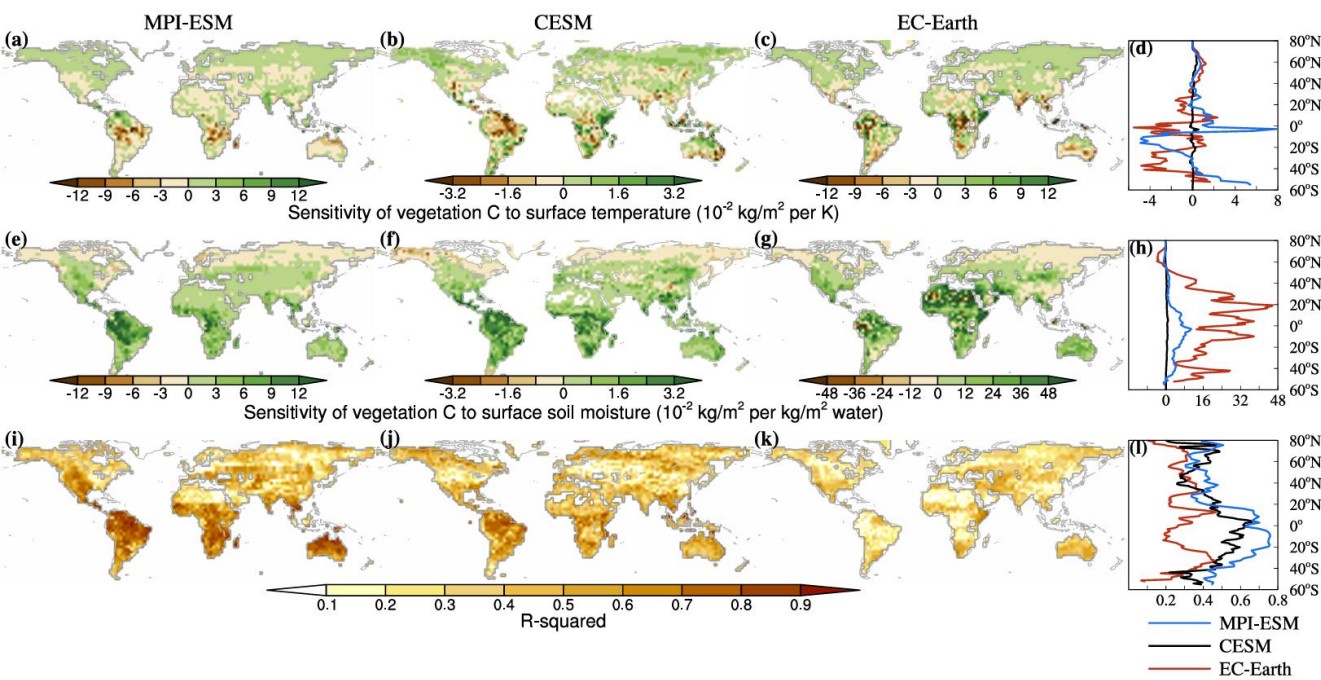

**Figure F3: Attribution of vegetation carbon changes to changes in near-surface air temperature (a-d) and surface soil moisture (e-h) and the respective R² values (i-l) for the irrigation of cropland scenarios (see Fig. 9 and F1 for cropland expansion and afforestation scenarios, respectively). For details see Fig. 9.**

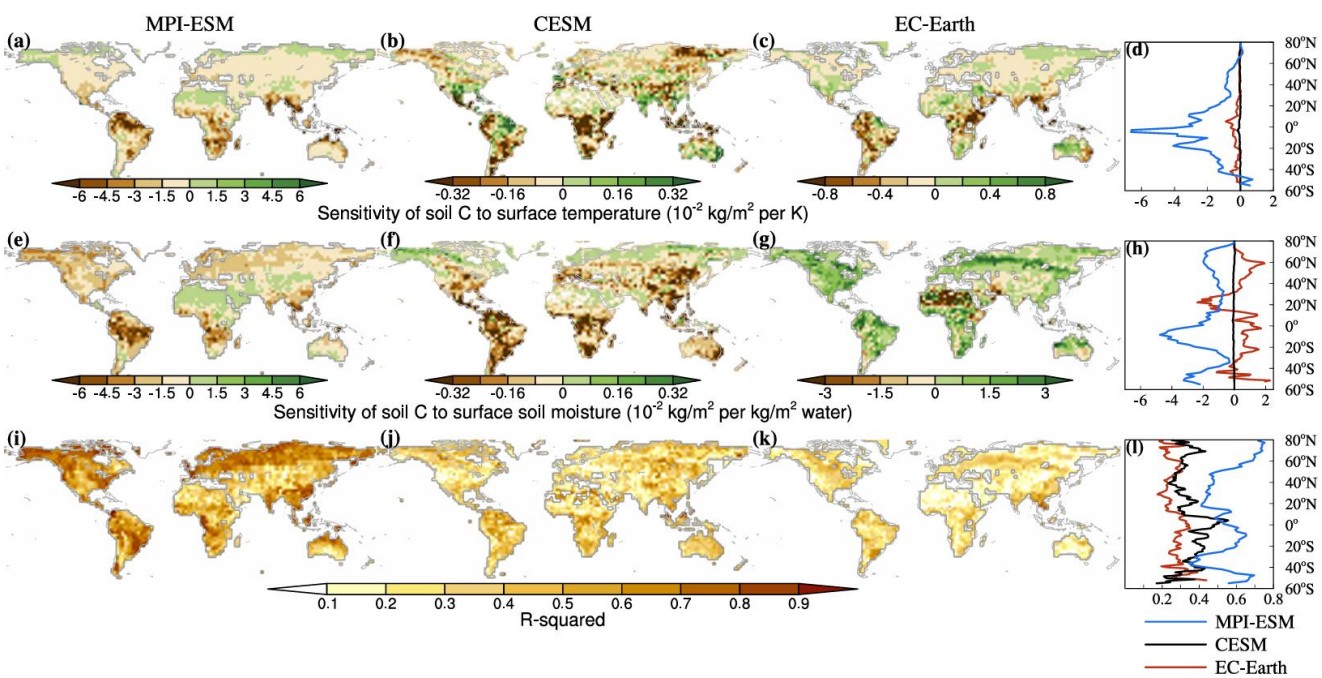

**Figure F4: Attribution of soil carbon changes to changes in near-surface air temperature (a-d) and surface soil moisture (e-h) and the respective R² values (i-l) for the irrigation of cropland scenarios (see Fig. 10 and F2 for cropland expansion and afforestation scenarios, respectively). For details see Fig. 9.**

**Code and data availability**

CESM is open source (https://www.cesm.ucar.edu/models/cesm2/release_download.html, last accessed: 26 July 2024). MPI-
ESM    is    available    under    the    MPI-M    software    license    agreement
(https://edmond.mpg.de/dataset.xhtml?persistentId=doi:10.17617/3.H44EN5, Model Development Team Max-Planck-Institut
für Meterologie, 2024). EC-Earth is available to institutes that have signed a memorandum of understanding with the EC-Earth
community and a software license agreement with the ECMWF. The source code can be requested from the EC-Earth
community via the EC-Earth website (http://www.ec-earth.org/, last accessed: 26 July 2024). The scripts used for data post-
processing and analysis can be found here: https://github.com/SuqiGuo/Guo_etal_2025_ESD (Guo, 2025). The data that
support    the    findings    of    this    study    can    be    found    here:    https://www.wdc-
climate.de/ui/entry?acronym=DKRZ_LTA_1147_ds00006 (Guo et al., 2024).

**Author contributions.**

The simulation protocol was designed by CFS, QL, WT, JP, FH, IM, SDH, and SG. SDH performed the simulations and produced the data using CESM. IM performed the simulations and produced the data using EC-EARTH. FH and SG performed the simulations and produced the data using MPI-ESM. SG analyzed the data and drafted the manuscript. TR implemented the irrigation scheme for MPI-ESM. HL assisted in setting up the MPI-ESM simulations. FH and SDH performed the post-processing for the signal separation. FL prepared the EC-EARTH data for post-processing and helped with the signal separation for EC-EARTH. DW and LN contributed to the explanation of signals in EC-EARTH. All authors commented on the paper and provided feedback on the data analysis.

**Conflict of interest statement**

The authors declare that they have no conflict of interest.

**Disclaimer**

Publisher's note: Copernicus Publications remains neutral with regard to jurisdictional claims in published maps and institutional affiliations.

**Acknowledgements**

This study was funded through the project "Land Management for Climate Mitigation and Adaptation" (LAMACLIMA). The LAMACLIMA project is part of AXIS, an ERA-NET initiated by JPI Climate, and funded by DLR/BMBF (DE, Grant No. 01LS1905A-B), BELSPO (BE, Grant No. B2/181/P1), NWO (NL, Grant No. 438.19.904), and co-funding by the European Union (Grant No. 776608). This work used resources of the Deutsches Klimarechenzentrum (DKRZ), granted by its Scientific Steering Committee (WLA) under Project ID bm1147. Resources for CESM simulations and data storage were provided by the VSC (Flemish Supercomputer Center), funded by the Research Foundation–Flanders (FWO) and the Flemish Government–department EWI. EC-EARTH simulations were conducted using platforms at the European Center for Medium-Range Weather Forecasts (ECMWF). SDH acknowledges funding by BELSPO (B2/223/P1/DAMOCO). FL acknowledges the VIDI award from the Netherlands Organization for Scientific Research (NWO) (Persistent Summer Extremes "PER SIST" project no. 016.Vidi.171.011). DW and LN were supported by the Strategic Research Area MERGE (ModElling the Regional and Global Earth system – www.merge.lu.se) and by the European Union's Horizon Europe research and innovation programme through the research project RESCUE (Grant No. 101056939). HL was supported by the European Union's Horizon 2020 research and innovation program through the research project 4C (Grant No. 821003) and the German Research Foundation (DFG) under Germany's Excellence Strategy the Cluster of Excellence "Climate, Climatic Change, and Society" (CLICCS) (Project No.

390683824). The authors would like to thank Björn Maier for his valuable suggestions and the icon design for Fig. 1, which greatly improved the visualization.

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
