# Peer review of "Remote carbon cycle changes are overlooked impacts of land-cover and land management changes"

_EGUsphere, 2024_

## Author Comment (AC1)

Dear Referee,

Thank you for your thorough and detailed review, which has been invaluable in enhancing the quality of our manuscript. We appreciate your recognition of the study's relevance, novelty, and extensive nature.

We have carefully considered each of your comments. Below, you will find our responses to the specific points, as well as the revisions we will make to the manuscript to address your suggestions.

Kind regards,
Suqi Guo and Co-authors
* * *
**Referee 1 Comment 1**

**This paper is very complicated to comprehend, primarily due to its structure and writing style. The descriptions of experiments and corresponding model simulations are tough to understand. There are a lot of differences among the different model experiments and their simulations. There are multiple variables considered and their interrelations. Consequently, there are multiple figures (seventeen, including the ones in the appendices, and each with multiple subplots!). However, these are not explained in sufficient detail. This makes the results inconclusive. Most of the interpretations are not clear and some of those are speculative in nature.**

**Response**

We appreciate the referee's comments regarding the clarity and structure of our manuscript. In response to the concerns raised, we will revise the text thoroughly to enhance the organization and readability. Below, we address each specific aspect mentioned and provide examples of our revisions.

1. **"The descriptions of experiments and corresponding model simulations are tough to understand."**

We recognize that the original text in Section 2.1 was difficult to comprehend. To improve clarity, we have paraphrased and simplified the language, particularly in Table 1 (now Table 2; see examples of revisions). We have also included a checkerboard pattern in Fig. C1 and linked it to the land cover forcing description in Section 2.1. To enhance the organization of this section, we have now introduced three scenarios together prior to discussing the model divergences in LCLMC implementations, which we have structured as points (i) to (iv). Additionally, as requested by another referee, we created a table summarizing these model divergences in setup. The complete revision will be included in the revised manuscript.

2. **"There are multiple variables considered and their interrelations. Consequently, there are multiple figures (seventeen, including the ones in the appendices, and each with multiple subplots!). However, these are not explained in sufficient detail. This makes the results inconclusive."**

To clarify how our analysis of several variables supports our research aims and to explain our variable choices, we have linked the Results sections to the research aims presented in the introduction (see lines 106-110). We have added a paragraph at the beginning of the Results section to introduce the variables used and the reason.

3. **"Most of the interpretations are not clear and some of those are speculative in nature."**

We have re-evaluated the speculative language used in our interpretations (e.g., "likely," "could," "may"). For claims well supported by evidence, we have removed these qualifiers. For instance, "is likely related to a cooling over the boreal latitudes" has been revised to "is related to a cooling over the boreal latitudes," as temperature change is indeed a driver of carbon stock changes (see line 318). Additionally, we removed the interpretation "In the case of MPI-ESM, this could be driven by warming (Fig. C2a), which increases both plant carbon assimilation and soil decomposition rates," due to insufficient evidence to support this conclusion (see line 333).

Further, we have rewritten parts of the interpretation sections and display items to enhance clarity:

**Examples of revisions in Section 2.1**

We have enhanced the readability and structure of the experimental design and the model divergences in the LCLMCs implementation.

Lines 146-164

[revised manuscript text omitted]

**Link between variable analysis and research aims**

lines 106-110

More specific aims of our study are (i) to quantify the simulated global development and spatial distribution of nonlocal effects of LCLMCs on different terrestrial carbon pools (Sects. 3.1 and 3.2), (ii) to assess the importance of nonlocal BGC effects in relation to the total effects, which consist of both local and nonlocal BGC effects and represent the overall carbon cycle response at the location of the LCLMCs (Sect. 3.3), (iii) the point in time when the nonlocal BGC effects become larger than the natural internal variability (Sect. 3.4), and (iv) the sensitivity of nonlocal BGC effects to temperature and soil moisture (Sect. 3.5).

**Variables' introduction**

First, we analyse the global-integral, transient carbon stocks changes in terrestrial carbon pools (Sect. 3.1). We then concentrate on particular carbon pools that influence total terrestrial carbon stock (cLand) changes as components. Specifically, we analyse changes in vegetation carbon stock (cVeg) and soil (cSoil), due to their dominance. Litter carbon stocks (cLitter) changes, being intermediate and temporal, often present similar but generally minor changes to cSoil (not shown). We also investigate the magnitude and importance of nonlocal BGC effects from both spatial distribution and relative magnitude perspectives (Sects. 3.2 and 3.3). Next, we investigate when these signals are established over time (Sect. 3.4). Lastly, we attribute the nonlocal BGC effects to climate factors, with climate distributions presented in Figs. C2 and C3 (Sect. 3.5).
* * *
**Referee 1 Comment 2**

**Perhaps it will be helpful for the readers to categorize the biogeophysical and biogeochemical effects into local and nonlocal categories and represent this information in a table.**
* * *
**Response**

We thank the referee for this valuable suggestion and agree that categorizing the biogeophysical and biogeochemical effects into local and nonlocal categories will enhance clarity for readers. In response, we have added a table that summarizes this information and linked it to the corresponding section in the introduction.

**Table 1: Definitions of Land-Cover and Land-Management Change (LCLMC) Effects (BGP: Biogeophysical, BGC: Biogeochemical).**

| LCLMCs effects | Affected regions | Definition |
|---|---|---|
| Local BGP effects | Regions with LCLMCs | LCLMCs influence the local climate via energy, water, and momentum fluxes due to changed land surface properties such as albedo, leaf area, and roughness. |
| Nonlocal BGP effects | Regions without LCLMCs | LCLMCs influence remote climate via advection of the altered air mass properties and possible changes in large-scale circulation. |
| Local BGC effects | Regions with LCLMCs | LCLMCs directly influence the local carbon emissions and sequestration. |
| Nonlocal BGC effects | Regions without LCLMCs | LCLMCs influence remote carbon stocks through climate changes driven by nonlocal BGP effects. |
* * *
**Referee 1 Comment 3**

**How the effects are separated into local and non-local is not objectively defined. Did they use any preset definition of the 'area of influence' or did they use a specific number of neighboring points (pixels) for such classifications?**

**Response**

We appreciate the referee's comment regarding the separation of local and nonlocal effects. We base our signal separation on the approach introduced by Winckler et al. (2017), where the processes are explained in detail (the illustration is provided in the response to Comment 9). We define changed and unchanged grid cells according to the land cover forcing applied in a checkerboard pattern, as clarified in Section 2.1 (Lines 148-151). Additionally, we have included the checkerboard pattern in Fig. C1 to aid in visualizing the setup. To further enhance clarity, we have reiterated this setup in Section 2.2 by adding the following sentence at the beginning:

> We simulate LCLMCs by applying land cover forcing in a checkerboard pattern of changed and unchanged land cover (Fig. C1).
* * *
**Referee 1 Comment 4**

**Lines 59-61: I do not totally understand this statement. Changing a forest to grassland will reduce its carbon sequestration, which will increase atmospheric warming. This is a global effect and biogeochemical in nature. On the other hand, an increase in albedo will impart a cooling that is local and biogeophysical in nature.**

**Response**

We appreciate the referee's comment and have revised the relevant section for clarity.

> The nonlocal effects can only be quantified by models. Studies changing forest to grasslands show that idealized deforestation, while it  can warm the climate on a global average with local BGP effects, brings about nonlocal BGP effects that cool the climate by several tenths of a degree on global average (Winckler et al., 2019a).

In this context, we focus on the impact of BGP effects on global climate, using this sentence as an introduction to nonlocal BGP effects. We refer to the global impact of local BGP effects to compare them with nonlocal BGP effects and emphasize the importance of nonlocal BGP effects on global mean temperature. While local BGP effects have substantial impacts on local climates, we can also derive a global integral of these effects, which is relevant for global-scale decision-making.

In the comment, there seems to be a misunderstanding regarding local and nonlocal BGC effects, possibly considering the global climate impact (including regions remote from LCLMCs) of biogeochemical effects. LCLMC-induced carbon sequestration and emissions—i.e., the biogeochemical effects of LCLMCs—always influence the global climate through GHG effects. However, we define local and nonlocal BGC effects based on whether the LCLMC-induced carbon pool changes occur locally or in regions remote from the LCLMCs. Local and nonlocal BGC effects are discussed in the following paragraphs. Definitions are also provided in Table 1 (see our response to Comment 2).

**Referee 1 Comment 5**

Lines 130-135: The second advantage of not considering the plausible realizations in the modeling experiments is not clear.

**Response**

We appreciate the referee's comment and have clarified the second advantage.

> Second, unlike historical LCLMCs or realistic future scenarios where LCLMCs occur in limited regions, idealized global LCLMCs enable the estimation and comparison of impacts across most regions worldwide.

**Referee 1 Comment 6**

CROP scenario in Table 1: So, here, basically, two land use types are considered: cropland and bare soil, right? And both these occupy 50% of each of the land grid cells? Are different biophysical and biogeochemical parameters used for the different crop types, such as albedo, rooting depth, etc.?

**Response**

We appreciate the referee's comments regarding the CROP scenario in Table 1. We recognize that the original text in the table was difficult to comprehend.

To clarify, the bare land fraction is maintained at the 2014 level, reflecting present-day climate conditions. Land cover change occurs only on hospitable land, and the 50% allocation refers to the global LCLMCs represented in a checkerboard pattern.

Regarding the specific plant functional types (PFTs) in the crop and forest categories, we have added the following paragraph at the beginning of Appendix A and linked it to Section 2.1 where PFTs are first mentioned:

Appendix A: Distribution of PFTs within crop and forest categories

> PFTs are used in earth system models to represent the diversity of land cover types within a grid box. These PFTs are grouped by biochemical and biophysical properties, which are represented by model-specific parameters. The number and specific types of PFTs within the crop and forest categories vary among Earth system models. MPI-ESM includes four forest types and one crop type. CESM includes eight forest types and nine crop types, with each crop type having a corresponding irrigated version for irrigation implementation. EC-Earth does not have specific forest types; instead, it uses a natural type that includes forests coexisting with grassland and shrubs, and it includes one crop type along with a corresponding irrigated version.

**Referee 1 Comment 7**

IRR scenario in Table 1: Since all the croplands were irrigated, did the authors check the hydrological budget and ensure its closure globally? Also, what kind of irrigation was considered, fed with surface or groundwater, drip or canal, etc.? These details are required.

**Response**

We thank the referee for this comment. MPI-ESM and CESM both use the flood irrigation method, while EC-Earth uses three methods: flood, sprinkler, and drip irrigation. MPI-ESM uses water from surface runoff and drainage, CESM uses water from rivers and oceans, and EC-Earth uses surface, reservoir, and upstream water. We have provided detailed information about the hydrological budget and irrigation methods in Appendix B. We have highlighted the relevant information for MPI-ESM and clarified the details for CESM and EC-Earth. Appendix B has been further linked to Section 2.1 for reader convenience (Line 159).

Appendix B (Lines 596-602)

> Regarding irrigation implementation, for the MPI-ESM, we adapted and implemented a simple irrigation scheme into JSBACH. It ==assures water mass conservation== in a coupled atmosphere/ocean climate model and maximizes the effect of irrigation to recycle locally available water to the atmosphere by evapotranspiration. ==Surface runoff and drainage== are first collected in a storage reservoir with 20 cm capacity before being transferred to the skin reservoir, filling it completely as long as water is available in the storage reservoir. In contrast to MPI-ESM, CESM and EC-Earth do not have a constraint on water availability. CESM applied daily

irrigation to the surface to retain a target soil moisture, while EC-Earth applied daily irrigation to the top of the soil column depending on the water deficit. While MPI-ESM and CESM use the flood irrigation method, EC-Earth incorporates three methods: flood, sprinkler, and drip irrigation. In CESM, irrigation water is first taken from river storage, with additional water drawn from the ocean when river water is insufficient. In EC-Earth, irrigation water is first taken from surface and reservoir water, with additional water drawn from neighbour cell surplus when local water is insufficient.

| Referee 1 Comment 8 |
|---|
| **Lines 160-162: See my earlier comment on the water budget.** |

**Response**

We appreciate the referee's comment. The details of the water budget are illustrated in Appendix B. For further clarification, please refer to our response to Comment 7.

| Referee 1 Comment 9 |
|---|
| **Section 2.2: The recipe used here to differentiate between the local and non-local effects is not clear. Can the authors show a flowchart summarizing these?** |

**Response**

We thank the referee for this suggestion. To clarify the approach used to separate local and non-local effects, we refer to the signal separation approach illustrated in Winckler et al. (2017), which provides a detailed step-by-step explanation. To avoid repetitive work, we prefer to cite this existing literature. Additionally, we include the relevant figure from Winckler et al. (2017) here for further reference:

[Figure]

**Fig. 1.**
Sketch illustrating the separation approach (arbitrary color scale). (a) The simulated signal. The LCC grid boxes stand out because there the signal (local plus nonlocal) is mostly stronger than in the surrounding non-LCC grid cells (only nonlocal). (b) The nonlocal effects at no-LCC boxes. (c) The nonlocal effects are interpolated to LCC boxes. (d) The difference at the LCC boxes between

the simulated signal in (a) and interpolated nonlocal effects in (c) is shown, which we then (e) interpolate in order to obtain global information on the local effects. This approach works analogously for extensive deforestation [(f)–(j)]. Grid boxes whose information is not used for interpolation in (b),(d),(g),(i) are shown in gray. (For results on local and nonlocal effects see Fig. 2.) Source: Winckler et al. (2017a).
* * *
**Referee 1 Comment 10**

**Lines 236-241: Why are changes in surface roughness due to land cover change and its impact on aerodynamic conductance not considered?**

**Response**

We appreciate the referee's comment. In Section 2.5 (lines 236-241), we focus on the direct climate drivers (nonlocal BGP effects) that influence nonlocal BGC effects. Changes in land physical properties, including albedo, roughness, and evapotranspiration, are indeed sources of both local and nonlocal BGP effects, as well as nonlocal BGC effects. While we consider nonlocal BGP effects—such as changes in temperature and moisture—we indirectly account for surface roughness. To isolate the specific contribution of roughness, simulations that explicitly impose changes in roughness are necessary to determine its impact on local and remote climate (Winckler et al., 2019), and subsequently, on the remote carbon cycle.

The definitions of local and nonlocal biogeophysical and biogeochemical effects are provided in the introduction section. To clarify, we reiterate them here for better understanding:

(i)Local biogeophysical effects arise from local LCLMCs, influencing the local climate through energy, water, and momentum fluxes due to altered land surface properties such as albedo, leaf area, and roughness.
(ii)Nonlocal biogeophysical effects occur when the advection of altered local air mass properties, combined with changes in large-scale circulation, affects remote climate.
(iii)Nonlocal biogeochemical effects result from changes in remote climate that further influence remote carbon stocks.
In Section 2.5, we analyze step (iii), acknowledging the potential and novelty of attributing nonlocal biogeochemical effects to specific surface physical properties.
* * *
**Referee 1 Comment 11**

**Figure 2: Please improve the description of different components of the terrestrial carbon cycle simulated by different models. In the present format, it is utterly confusing for the readers. For example, what about the blue dashed line? To interpret this the reader has to read the caption. Why then some other lines are described in the figure itself? Use a uniform and detailed description and present those in a way straightforward to understand.**

**Response**

We appreciate the referee's feedback and have revised the legend of Figure 2 to improve clarity. Specifically, we have provided separate legends by presenting the carbon pools in different colors and the models in different line patterns. Additionally, we have changed the line pattern color in the legend for models to a neutral light grey to avoid confusion. The usage of colors and patterns is uniform through all the subplots in the figure, for instance, the blue dashed lines show time series of the litter carbon pool from CESM simulations in all the subplots. We also revised the caption of Figure 2 for further clarification.

[Figure]

Figure 2

[Figure]

**Figure 2: Simulated nonlocal effect on the development of global terrestrial carbon pools after an idealized change of 50 % of all grid cells: (a)  cropland expansion (b) afforestation (c) irrigation of cropland expansion. The total land carbon (black) is separated into vegetation (green), soil (orange), and litter (blue) pools. The results for MPI-ESM (solid lines), CESM (dashed lines), and EC-Earth (dotted lines) are shown for each carbon pool and for total land carbon.**

| Referee 1 Comment 12 |
| --- |
| **Figure 3: How are the boxes selected in panel (a) for calculating the regional averages? The boxes over northern America and Australia include oceans as well, which should not be counted in the terrestrial carbon cycle. Why don't the authors use appropriate shapefiles to select and crop the regions of their interest?** |

**Response**

The regions were selected based on three reasons: they show a strong nonlocal signal, consistency across models, and represent a range of latitudes. We clarified the reasons with the following revision:

> We aggregate results to a few core regions. These regions were chosen because they exhibit a large absolute nonlocal signal and the signal across models is consistent (Fig. 5, 6). Additionally, we choose regions across various latitudes to capture latitudinal diversity.

The inclusion of ocean areas in the selected regions does not affect our results, as we calculate the total terrestrial carbon, which is zero over oceans. While this approach might lead to differences if we were analyzing carbon density changes normalized by area, it remains accurate for our purposes. To address

any potential confusion, we have revised the caption for Fig. 5 as follows:

> **Figure 5: Relative contribution of the nonlocal to total effect of LCLMC on vegetation carbon of the last 30 years in the 160 year simulation period using MPI-ESM (green), CESM (red), and EC-Earth (orange) for (a) cropland expansion, (b) afforestation, and (c) irrigation of cropland expansion. Values are separated into the global integral (global) and regional means for Eastern North America  (ENA),  Western Amazon (WAAM), Central Congo Basin (CCB), Northern Eurasia (NE), Northeastern Asia  (NEA), Southern Southeast Asia  (SEA), and Northern Australia (NAU). Regions are defined within the red boxes in Fig. 3a, and only terrestrial areas are considered. **
* * *
**Referee 1 Comment 13**

**Can the authors show spatial trends of different carbon pools for different experiments and models over the simulation period for the aforementioned regions?**
* * *
**Response**

We thank the referee for this valuable suggestion. We interpret it as having two possible meanings: (1) displaying time series of spatially averaged values, or (2) presenting global maps that illustrate the trends of carbon pools (calculated as the slope of linear regressions). We find both interpretations to be valuable.

However, given that the carbon pools do not change linearly over the simulation period and instead show saturating trends in some cases across models and scenarios, we have opted to focus on time series for the selected regions. We have created figures showing the development of different carbon pools in the selected regions, which are added in Appendix D. These figures helped us gain a better understanding of the global integrated nonlocal BGC signal in Sect. 3.1 and the time of emergence in Sect. 3.4, and we have integrated the relevant discussion into the main text.

**Example figure**

**Appendix D: Temporal development of regionally integrated nonlocal BGC effects**

[Figure]

**Figure D1: Simulated nonlocal effect on the development of terrestrial carbon pools in the Western Amazon after an idealized change of 50 % of all grid cells: (a) cropland expansion (b) afforestation (c) irrigation of cropland expansion. The total land carbon (black) is separated into vegetation (green), soil (orange), and litter (blue) pools. The results for MPI-ESM (solid lines), CESM (dashed lines), and EC-Earth (dotted lines) are shown for each carbon pool and for total land carbon.**

**Relevant discussion:**

Lines 281-283

> Additionally, the nonlocal BGC stock changes show strong interannual variability, particularly in MPI-ESM and EC-Earth, which can be related to internal climate variability (Loughran et al., 2021). This variability is primarily driven by the Western Amazon region, while the Central Congo Basin shows much weaker variability (see Appendix D for details). This variability is related to the internal climate variability, which varies across regions (Loughran et al., 2021).

Lines 380-385

> The mid-latitudes show a later ToE, with different magnitudes across models. For example, Eastern North AmericaNorth America typically shows a later ToE for all three models while North Eurasia also shows a late ToE for CESM and EC-Earthwhich is indicated by the relatively flat trend in the temporal development of carbon pools during the initial decades (Figs. E1 and D4). These regions are primarily characterized by Crop- and grasslands take a considerable fraction of land in Eastern North America, indicating that the response of those land cover types is slower than that of forests. However, for MPI-ESM and CESM, the nonlocal BGC effect in Eastern North AmericaNorth America reaches a magnitude similar comparable to that in Northern EurasiaNorth Eurasia, East Asia Northeastern Asia, and Southern Southeast Asia Southeast Asia by the end of the simulation period (see Fig. 3 and Appendix D for details). This suggests that the nonlocal climate impact on crop- and grasslands persistently accumulates over

time, and ultimately becomes comparable to that on forests.
* * *
**Referee 1 Comment 14**

**Consistent scales should be used across the subplots in all the figures. For example, the scales on the y-axis are different in the subplots of Figure 2. Similarly, the x-axis scales are different in panels (d), (h) and (l) in Figure 3. This makes the intercomparison difficult.**

**Response**
We thank the referee for this valuable suggestion. Regarding Figure 2, we realize that the current presentation may cause some confusion. The y-axis scales are actually consistent across the subplots in Figure 2. However, to reduce confusion, we have now added the same top and bottom values for the y-axis in panels (b) and (c), which were initially removed to maintain a clean and compact look since these panels do not span the full range of values as displayed in panel (a) (see Figure 2 in our response to Comment 11)

For Figures 3 and 4, we have made the x-axis scales consistent in panels (d), (h), and (l) to improve comparability. However, in Figures 5 and 6, it is challenging to use a consistent y-axis scale across all panels due to the substantial difference in magnitude among different scenarios. Consistent scales would render some signals nearly invisible. A similar issue applies to Figures 9 and 10, where large differences in magnitude—ranging from several times to an entire order of magnitude—make consistent scaling impractical for both the spatial maps and latitudinal mean plots. For Figures 7 and 8, we have already employed consistent scales across all subplots.

Figure 3:

[Figure]

Figure 4:

[Figure]

**Referee 1 Comment 15**

**Lines 379-383: Shouldn't the ToE for the forested regions should be slower than croplands and grasslands, since the total biomass content is significantly higher in the former land use type?**

**Response**

We thank the referee for this comment. Our simulations indicate that the ToE for forested regions occurs earlier than in cropland and grassland regions. We explained this in the Discussion section (lines 494-497), where we link to the Methods section for the relevant definitions of noise and signal for clarity.

Based on these definitions, we explain this signal in detail as follows: we define noise as the variability of carbon pools in the CTL scenario, which reflects internal natural variability. Forested regions exhibit higher noise due to their high biomass density, making them more sensitive to internal climate variability. High biomass density also amplifies the signals, representing the response of the carbon cycle to nonlocal climate changes. The amplification is stronger for signals that consider larger nonlocal climate changes relative to natural variability. Additionally, forested regions experience more substantial climate signals in the CROP scenario, further accelerating the ToE compared to other land-use types.

Line 494-497:

> In addition, dense forests experience an earlier ToE than other types. One reason is the higher sensitivity of carbon pools in these regions to nonlocal BGP effects (Fig. 9 and 10), which is caused by the high biomass density of the forest. For the CROP scenario, the transition from forest to cropland in the Amazon and Congo region (Fig. C1) causes substantial nonlocal BGP effects on nearby regions.

**Referee 1 Comment 16**

**Figure C1: What do the panels j and k stand for? This should be described in the figure caption.**

**Response**

We appreciate the referee's observation. The subplots j and k, which were initially included for the wood harvest scenario, have been removed as they are not discussed in the manuscript.

**Referee 1 Comment 17**

**Several references cited in the text are missing from the bibliography, such as Arora et al.**

**Response**

We thank the referee for this observation. We confirm that Arora et al. is indeed cited in the reference list (lines 677–679). Upon reviewing the manuscript, we also identified a missing reference, which has now been added as follows:

> Loughran, T. F., Boysen, L., Bastos, A., Hartung, K., Havermann, F., Li, H., Nabel, J. E. M. S., Obermeier, W. A., and Pongratz, J.: Past and future climate variability uncertainties in the global carbon budget using the MPI Grand Ensemble, Global Biogeochem. Cy., 35, e2021GB007019, https://doi.org/10.1029/2021GB007019, 2021.

**Referee 1 Comment 18**

**Technical corrections:**
1. **Line 165: "are shown" instead of "is shown".**
2. **Figure 5: CESM is marked blue; however, in the caption, it is written as red.**
3. **Lines 467-483: This is a big paragraph. Consider breaking it into two or more.**

**Response**

1. We have revised Line 165 as follows:

> The global distributions of land-cover changes and magnitude of irrigation application  are shown in Fig. C1.

2. We have revised the caption of Figure 5 as follows:

**Figure 5: Relative contribution of the nonlocal to total effect of LCLMC on vegetation carbon of the last 30 years in the 160 year simulation period using MPI-ESM (green), CESM (red), and EC-Earth (orange) for (a) cropland expansion, (b) afforestation, and (c) irrigation of cropland expansion. Values are separated into the global integral (global) and regional means for Eastern North America  (ENA),  Western Amazon (WAAM), Central Congo Basin (CG), Northern Eurasia (NE), Northeastern Asia  (NEA), Southern Southeast Asia  (SSEA), and Northern Australia (NAU). Regions are defined within the red boxes in Fig. 3a, and only terrestrial areas are considered. **

3. We have broken the paragraph into two shorter paragraphs for readability.

---

## Author Comment (AC2)

Dear Referee,

First of all, thank you for taking your valuable time to review our manuscript which we want to publish in the journal Earth System Dynamics. We are thankful for your generally very positive feedback, that you judge our study as a timely effort and that you share the opinion that local-nonlocal BGC effects of LCLMCs are so far neglected in scientific literature. We also hope to stimulate further investigations in this field.

We carefully went through your comments (published here https://doi.org/10.5194/egusphere-2024-2387-RC2) and hope that we could address all the suggestions, issues and concerns that you raised in your referee comment.

Below you find our responses to the specific points you raised, if needed, additionally with the corresponding text paragraph and how we changed it in order to address your suggestions or concerns.

Thank you again for your valuable input,
Kindest regards,
Suqi Guo and co-authors
* * *
**Referee 2 Comment 1**

**Although this is a first attempt to study the nonlocal BGC effects using model sensitivity simulations, a model ensemble involving multiple ESMs, would have been more appropriate for the study. Model ensemble mean and spread would add more insights into the effects of LCLMCs. Authors can consider this aspect.**

**Response**

We appreciate the referee's suggestion regarding the use of a model ensemble mean and spread for added insight. While this approach is valuable for a large number of models, the current study involves only three models with a relatively large spread in their responses. Given the small ensemble size, calculating a robust and representative ensemble mean is challenging and may not yield reliable results. Therefore, we would rather investigate individual model results and in this way the common features and differences among the three models can be identified for further attribution and process understanding.
* * *
**Referee 2 Comment 2**

**There are noticeable differences in the simulated nonlocal BGC effects among three ESMs, reflected in spatial maps, latitudinal means, and sub-regions. This is expected; however, separate sub-sections within "Results" sections should be devoted to explaining the differences (on each aspect considered)**

**Response**

Thank you for your suggestion. We will restructure and include a dedicated paragraph in each results section to explain the differences among models. For example, in Sect. 3.1, we addressed the divergence in model responses regarding the magnitude, trend, and variability of the global-integral nonlocal BGC effects.

> There are several reasons for the divergence across models regarding the magnitude, trend and variability of global-integral, transient nonlocal BGC effects. First, the magnitude divergence is dominant by some key regions where nonlocal BGP effects diverge considerably (see Sect. 3.2). For example, for the CROP scenario,  opposing nonlocal Veg and cLand changes between MPI-ESM/CESM and EC-Earth are mainly caused by opposing cVeg changes in the Western Amazon  region due to opposing nonlocal climate conditions (see Sect. 3.2.1,  Sect. In EC-Earth,thein EC-Earth between aBGC stock gain and losscan be attributed to the dynamic vegetation competition and replacement, as well as the gradual establishment of tree physical properties.~~

| Referee 2 Comment 3 |
|---|
| **The whole analysis of the sensitivity simulations can be improved by avoiding multiple spatial maps and focusing on latitudinal means (e.g., Figure 3 d, h, l) and sub-regional contributions (e.g., Figure 5, 6).** |

**Response**

We appreciate the referee's suggestion. We have moved the spatial maps from Figures 7 and 8 to the Appendix and revised Section 3.4 to focus on the latitudinal means. However, we chose to retain the other maps, as they enhance our analysis and provide a more comprehensive perspective.

Specifically, the spatial maps in Figures 3, 4, 9, and 10 offer an intuitive overview of the signal-dominant regions or those most sensitive to climate factors within each latitude band. These regions correlate with the initial PFT distributions, such as forests, grasslands, or croplands. This spatial context can be lost or less apparent when presenting only latitudinal means and sub-regional contributions.

Additionally, the spatial maps in Figures 3 and 4 clarify interactions among regions by illustrating how signals either enhance or offset across areas. This comprehensive spatial perspective helps to reveal regional relationships and interactions that may otherwise remain obscured. We believe that the present arrangement of figures keeps a good balance of showing integrated information with the latitudinal and regional means and more details of regional features and distributions with spatial maps.

Figures 7 and 8

[Figure]

**Figure 7: Latitudinal mean time of emergence for nonlocal vegetation carbon changes surpassing natural variability in the (a) cropland expansion, (b)afforestation, and (c) irrigation of cropland expansion scenario (c). Results are shown for MPI-ESM (blue), CESM (black), and EC-Earth (red).**

[Figure]

**Figure 8: Latitudinal mean time of emergence for nonlocal soil carbon changes surpassing natural variability. For details see Fig. 7 | Same as Fig. 7 but for soil carbon.**

**3.4 Time of emergence**

Generally, nonlocal cVeg changes emerge within less than 40 years (Figs. 7 and E1) for the majority of the hospitable land area for all LCLMC scenarios. The latitudinal mean ToE shows a similar pattern of variation with latitude across models and scenarios. In the tropics and Northern Hemisphere high latitudes, the ToE occurs earlier, typically within 30 years. In the tropics, this early ToE is dominated by  the Western Amazon and  Central Congo Basin, i.e. for rather forested regions, with ToE could be even shorter than ten years, depending on the model and scenario (Fig. E1). The mid-latitudes show a later ToE, with different magnitudes across models. For example, Eastern North America typically show a late ToE which is indicated by the relatively flat trend in the temporal development of carbon pools during the initial decades (Figs. E1 and D4). Crop- and grasslands take a considerable fraction of land in Eastern North America, indicating that the response of those land cover types is slower than that of forests. However, for MPI-ESM and CESM, the nonlocal BGC effect in Eastern North America reaches a magnitude comparable to that in Northern Eurasia,  Northeastern Asia, and  Southeast Asia  by the end of the simulation period ( Fig. 3 and Appendix D for details). This suggests that the nonlocal climate impact on crop- and grasslands persistently accumulates over time, and ultimately becomes comparable to that on forests.

~~Similarly, for FRST and IRR scenarios, the ToE is shortest in regions with largest nonlocal cVeg changes by the end of the simulation period. This comprises small regions within the Congo and the Amazon, and the North Eurasia region for CESM and EC-Earth. EC-Earth generally shows a large magnitude of nonlocal cVeg changes in the Amazon and Congo regions for all scenarios. However, the ToE is generally larger than in MPI-ESM and CESM. The reason could again be the effect of the dynamic vegetation competition and replacement. Additionally, for the FRST scenario, gradual establishment of tree physical properties delays the growth trend and ToE.~~

For cSoil, the ToE is also generally shorter than 40 years for the majority of the hospitable land area for all scenarios and models (Figs. 8 and E2). The latitudinal mean ToE shows smaller variation for all models and scenarios. The ToE for cSoil is typically shorter than that for cVeg, which is related to the relatively smaller internal variability of cSoil ~~In most cases, the ToE is shorter in regions with large nonlocal cSoil changes, for example: the Amazon and Congo regions in MPI-ESM and EC-Earth for the CROP scenario; the Congo region in MPI-ESM for the FRST scenario; the North America region in MPI-ESM and CESM for the IRR scenario. In contrast, for EC-Earth, even though nonlocal cSoil changes are smaller than nonlocal cVeg changes in key regions like the Amazon, Congo, and North Eurasia, the ToE is typically shorter. This could be due to the relatively smaller internal variability of cSoil.~~
* * *
**Referee 2 Comment 4**

**The abstract and conclusion sections should contain some quantitative statements: the magnitude of nonlocal BGC compared to total effects (%) on different regions, impacts of temperature on BGC effects, etc.**

**Response**

We appreciate the referee's suggestion. We already show quantitative data for time of emergence, total BGC effects globally and region specific. We now additionally include the nonlocal to total BGC effects ratio in both the abstract and conclusion sections. Additionally, in the conclusion section, we add quantitative data for the sensitivity of carbon cycle components to changes in climate variables. To avoid overloading the abstract with details, we focus on the overall importance of nonlocal BGC effects in the abstract, while including specific sensitivities in the conclusion for those interested in further attribution analysis.

**Revised sentence in the abstract (Lines 34-35):**

For the irrigation scenario, the nonlocal BGC effects are comparable to the total BGC effects with the nonlocal-to-total ratio for vegetation carbon pools commonly reaching around 90%.

**Revised sentences in the conclusion (Line 566):**

For the IRR scenario, the nonlocal BGC effects are typically comparable or exceed the total effects with the nonlocal to total ratio for vegetation carbon pools commonly reaching around 90%. Nonlocal BGC effects can be attributed to nonlocal climate changes such as changes in temperature and soil moisture, with tropical regions being particularly sensitive. In these regions, every Kelvin increase in temperature results in a decrease of over 10 GtC in cVeg. The cVeg response to soil moisture changes varies across models, with each millimeter increase in soil moisture leading to a rise in cVeg of +85 to more than +200 GtC.
* * *
**Referee 2 Comment 5**

**The definition and discussion of local versus non-local and BGP versus BGC were described in detail in the introduction section. The first 4-5 sentences of the abstract can be rewritten to comprehend these aspects concisely. An additional table (in addition to Figure 1; a nice figure!) can be included and discussed in the introduction section itself to make these definitions clearer to the readers.**

**Response**

We thank the referee for these valuable suggestions. We have rewritten the considered sentences of the abstract the following:

Land-cover and land management changes (LCLMCs) have a substantial impact on the global carbon budget and, consequently, global climate via the biogeochemical (BGC) effects. The commonly considered BGC effects refer to the direct influence of LCLMCs on local carbon stocks, namely the local BGC effects. However, LCLMCs also influence climate by altering the local surface energy balance due to changes in land surface properties such as albedo, leaf area, and roughness, namely local biogeophysical (BGP) effects. Altered local air mass properties can impact regions remote from LCLMCs through advection and changes in large-scale circulation, namely nonlocal BGP effects. BGP effects act locally, but also nonlocally through advection or atmospheric circulation changes. Previous studies have shown potentially substantial nonlocal BGP effects on temperature and precipitation. Given that the terrestrial carbon cycle strongly depends on climate conditions, this raises the question of whether LCLMCs can trigger remote carbon cycle changes, namely the nonlocal BGC effects - a currently overlooked potentially large climate and ecosystem impact. To assess these nonlocal biogeochemical (BGC) effectsthe nonlocal BGC effects, we analyze sensitivity simulations for three selected types of hypothetical large-scale LCLMCs: global cropland expansion, global cropland expansion with irrigation, and global afforestation, which were performed by three state-of-the-art Earth system models.

Additionally, we have added the following table to clarify the definitions of local versus nonlocal and BGP versus BGC effects.

**Table 1: Definitions of Land-Cover and Land-Management Change (LCLMC) Effects (BGP: Biogeophysical, BGC: Biogeochemical).**

| LCLMCs effects | Affected regions | Definition |
|---|---|---|
| Local BGP effects | Regions with LCLMCs | LCLMCs directly influence the local climate via energy, water, and momentum fluxes due to changed land surface properties such as albedo, leaf area, and roughness. |
| Nonlocal BGP effects | Regions without LCLMCs | LCLMCs influence the climate of remote regions via advection of the altered air mass properties and possible changes in large-scale circulation. |
| Local BGC effects | Regions with LCLMCs | LCLMCs directly influence the local carbon emissions and sequestration by changing the local vegetation type or its management. |
| Nonlocal BGC effects | Regions without LCLMCs | LCLMCs influence the carbon stocks of remote regions through climate changes driven by nonlocal BGP effects. |
* * *
**Referee 2 Comment 6**

**The specifics of the three models (and their differences, e.g., use of LPJ in EC-Earth) are explained in detail in the methods section (Sect. 2.1). It would be better if the authors added a table listing three model details that are important for the analysis in this study.**

**Response**

We thank the referee for this helpful suggestion. We have added the following table to Section 2.1 to provide a concise overview of the key differences among the three models regarding LCLMCs

implementation.

**Table 2: Comparison of land-cover and land management changes (LCLMCs) implementations across Earth system models (PFT: plant functional type).**

| LCLMCs implementation | MPI-ESM | CESM | EC-Earth |
|---|---|---|---|
| Land surface scheme | JSBACH3 | CLM5 | LPJ-GUESS |
| Dynamic vegetation | Dynamic competition among PFTs switched off | Dynamic competition among PFTs switched off | Dynamic competition among six stand types (natural, pasture, urban, crop, irrigated crop, peatland). |
| Afforestation implementation | Uses prescribed transitions to model intrinsic forest PFTs. | Uses prescribed land cover states for model intrinsic forest PFTs. | Does not support exact afforestation fractions; afforestation occurs by expanding natural PFT, allowing coexistence of grassland and shrubs. |
| Plant physical properties establishment | Immediate establishment after land cover change | Immediate establishment after land cover change | Gradual establishment based on biomass accumulation. |
| Water cycle coupling | Fully coupled between land and atmosphere | Fully coupled between land and atmosphere | Uncoupled to atmospheric model; irrigation has no direct impact on atmosphere (e.g., through surface water and energy fluxes). |

**Referee 2 Comment 7**

**Better assigning different colors to different models in Figure 3 d, h, l (latitudinal means) to distinguish between color scales in the spatial maps.**

**Response**

We appreciate the referee's suggestion, which highlights the potential confusion caused by the similar color schemes. In the latitudinal mean plots, we have now employed distinct colors different from those used in the spatial maps to enhance clarity.

Figure 3:

[Figure]

**Referee 2 Comment 8**

**The naming of different regions (Figures 3a and 5) should be more careful: North America to Eastern NA, Northern Australia, etc**

**Response**

We appreciate the referee's attention to the accuracy of regional names. We have revised the names to be

more geographically descriptive as follows: Eastern North America (ENA), Western Amazon (WA), Central Congo Basin (CCB), Northern Eurasia (NE), Northeastern Asia (NEA), Southern Southeast Asia (SSEA), and Northern Australia (NAU). These updated names have been consistently applied throughout the manuscript.

| Referee 2 Comment 9 |
| --- |
| **It is better if the authors are consistent while describing the three scenarios in the following order: CROP, IRR, and FRST, throughout the manuscript.** |

**Response**
We agree that maintaining consistency in describing the three scenarios enhances the organization of the manuscript. We have reorganized the order of the scenarios to follow the suggested sequence: CROP, IRR, and FRST throughout the text.

---

## Author Response (AR2)

**Remote carbon cycle changes are overlooked impacts of land-cover and land management changes**

*Response to referee 1*

We sincerely thank the referee for their thoughtful review of our manuscript. We apologize for the confusion caused by the line number references in our previous response. In reply to comment 3, we had indicated Lines 162-166 in the *Author's tracked changes* document (egusphere-2024-2387-ATC.pdf) for the land forcing information. However, we realize we did not clarify this reference in our earlier response, and we apologize for the oversight.

For clarity, we now continue to refer to the line numbers in the current version of the *Author's tracked changes* document for this round of revisions.

We appreciate your understanding and the opportunity to improve the manuscript.

| Referee 1 Comment 1 |
| --- |
| The response to my previous comment No. 3 is not clear. This question was regarding the spatial scale of the local and non-local effects. In other words, I wanted to know the effects of big areas being considered local and non-local. The authors can either state in in terms of area (km2) or number of pixels or grid points. It is required but not enough to refer to an old study. These need to be explained step-by-step or included in a flowchart. |

**Response**

We apologize for not fully understanding the requested modifications to the manuscript in our previous response and now provide more information on the scale of local and non-local areas and effects.

We have added the information in terms of area by providing the modeling resolution to Tab. 3. It should be noted that our study does not investigate how far-reaching non-local effects in a specific case are. Such aspects have been addressed by a similar author team looking into 'length scales' of moisture recycling (De Hertog et al., 2024). Instead, our study compares the entirety of non-local effects to those happening at the location of LCLMC. That the distinction between local and non-local areas and effects depends on the application – with the global focus, and thus ESM resolution, in our case – has now been clarified in the introduction by adding the sentence:

> The definition of local vs nonlocal scales depends on the application – while changes in micro- or mesoscale phenomena could be resolved by high-resolution modelling, our study focuses on global impacts of large-scale LCLMCs connected to synoptic scales.

And in the technical implementation in Sec. 2.2, where we reference the new information of Tab. 3:

> As we are interested in global impacts of large-scale LCLMCs at the scale of 100 km upwards, which matches the resolution of the ESMs, we implement the checkerboard pattern at the native resolution of each ESM (Tab. 3).

Additionally, we have accommodated the reviewers' suggestion for a clear step-by-step description for the isolation of the local and nonlocal effects in Sec. 2.2, now explicitly including the information from the referenced literature, even for the first steps where we apply the method in an unaltered way. This

should greatly improve clarity.

<table>
<tr><td>Referee 1 Comment 2</td></tr>
<tr><td>My previous comment No. 15: The response of the authors to this query is not clear. Specifically, the authors state that forests exhibit a larger internal natural variability because of their larger biomass content. I am not sure how these two are linked.</td></tr>
</table>

**Response**

We thank referee for the insightful feedback.

In our study, we define internal natural variability (or noise) as fluctuations in carbon pools driven by natural climate variability. Forests, with their higher biomass density, are more sensitive to these climate fluctuations. For example, when climate conditions become unfavorable (e.g., during droughts), the higher biomass density in forests can lead to higher carbon emissions due to increased carbon decomposition and tree mortality. Conversely, when the climate becomes more favorable, these larger biomasses sequester more carbon, leading to greater carbon uptake. Thus, forests exhibit higher internal variability of carbon pools compared to croplands or grasslands due to their higher biomass density.

However, while forests show larger internal variability (noise), it is not the dominant factor influencing the Time of Emergence (ToE) in forested regions. The earlier ToE is primarily driven by the signal—the amplified response of the carbon cycle to non-local BGP climate changes. In forests, the higher biomass density intensifies this signal, making the response to non-local climate changes more pronounced compared to other land-use types. This stronger signal accelerates the ToE, despite the larger internal variability.

In summary, although the higher biomass density in forests leads to larger internal variability, it is the amplification of the signal due to non-local BGP climate effects that dominantly drives the earlier ToE in forested regions. We have added a sentence in the Sec. 4.1 to clarify this to the reader.